

# Illustration of microphysical processes in Amazonian deep convective clouds in the Gamma phase space: Introduction and potential applications

Micael A. Cecchini[1], Luiz A. T. Machado[1], Manfred Wendisch[2], Anja Costa[3], Martina Krämer[3], Meinrat O. Andreae[4,5], Armin Afchine[3], Rachel I. Albrecht[6], Paulo Artaxo[7], Stephan Borrmann[4,8], Daniel Fütterer[9], Thomas Klimach[4], Christoph Mahnke[4,8], Scot T. Martin[10], Andreas Minikin[11], Sergej Molleker[8], Lianet H. Pardo[1], Christopher Pöhlker[4], Mira L. Pöhlker[4], Ulrich Pöschl[4], Daniel Rosenfeld[12], Bernadett Weinzierl[9,13,14]

[1]Centro de Previsão de Tempo e Estudos Climáticos, Instituto Nacional de Pesquisas Espaciais, Cachoeira Paulista, Brasil.
[2]Leipziger Institut für Meteorologie (LIM), Universität Leipzig, Stephanstr. 3, 04103 Leipzig, Deutschland.
[3]Forschungszentrum Jülich, Institut für Energie und Klimaforschung (IEK-7), Jülich, Germany.
[4]Biogeochemistry, Multiphase Chemistry, and Particle Chemistry Departments, Max Planck Institute for Chemistry, P.O. Box 3060, 55020, Mainz, Germany.
[5]Scripps Institution of Oceanography, University of California San Diego, La Jolla, CA 92037, USA.
[6]Departamento de Ciências Atmosféricas, Instituto de Astronomia, Geofísica e Ciências Atmosféricas (IAG), Universidade de São Paulo (USP), Brasil.
[7]Instituto de Física (IF), Universidade de São Paulo (USP), São Paulo, Brasil.
[8]Institut für Physik der Atmosphäre (IPA), Johannes Gutenberg-Universität, Mainz, Deutschland.
[9]Institut für Physik der Atmosphäre, Deutsches Zentrum für Luft- und Raumfahrt (DLR), Oberpfaffenhofen, 82234 Wessling, Deutschland.
[10]School of Engineering and Applied Sciences and Department of Earth and Planetary Sciences, Harvard University, Cambridge, Massachusetts, USA.
[11]Flugexperimente, Deutsches Zentrum für Luft- und Raumfahrt (DLR), Oberpfaffenhofen, Deutschland.
[12]Institute of Earth Sciences, The Hebrew University of Jerusalem, Israel.
[13]Faculty of Physics, University of Vienna, Boltzmanngasse 5, 1090 Wien, Austria.
[14]Ludwig-Maximilians-Universität, Meteorologisches Institut, München, Deutschland.

*Correspondence to*: M. A. Cecchini (micael.cecchini@gmail.com)

**Abstract.** The behavior of tropical clouds remains a major open scientific question, given that the associated physics is not well represented by models. One challenge is to realistically reproduce cloud droplet size distributions (DSD) and their evolution over time and space. Many applications, not limited to models, use the Gamma function to represent DSDs. However, there is almost no study dedicated to understanding the phase space of this function, which is given by the three parameters that define the DSD intercept, shape, and curvature. Gamma phase space may provide a common framework for parameteri-




zations and inter-comparisons. Here, we introduce the phase-space approach and its characteristics, focusing on warm-phase microphysical cloud properties and the transition to the mixed-phase layer. We show that trajectories in this phase space can represent DSD evolution and can be related to growth processes. Condensational and collisional growth may be interpreted as pseudo-forces that induce displacements in opposite directions within the phase space. The actually observed movements in the phase space are a result of the combination of such pseudo-forces. Additionally, aerosol effects can be evaluated given their significant impact on DSDs. The DSDs associated with liquid droplets that favor cloud glaciation can be delimited in the phase space, which can help models to adequately predict the transition to the mixed phase. We also consider possible ways to constrain the DSD in two-moment bulk microphysics schemes, where the relative dispersion parameter of the DSD can play a significant role. Overall, the Gamma phase-space approach can be an invaluable tool for studying cloud microphysical evolution and can be readily applied in many scenarios that rely on Gamma DSDs.

## 1 Introduction

Tropical deep convective clouds (DCCs) constitute an important source of precipitation (Liu 2011) and they interact with atmospheric solar and terrestrial radiation, dynamical processes and the hydrological cycle (Arakawa 2004). Deep tropical convection is responsible for transporting energy upwards and thus sustaining the Hadley circulation that redistributes heat to higher latitudes (Riehl and Malkus 1958; Riehl and Simpson 1979; Fierro et al. 2009, 2012). Therefore, understanding the processes that impact the characteristics of tropical DCCs is crucial in order to comprehend and model the Earth's climate.

The DCCs over the Amazon are of particular interest. Given the homogeneity of the surface and the pristine air over undisturbed portions of the rainforest, Amazonian DCCs can have similar properties to maritime systems (Andreae et al. 2004). At the same time, their daily persistence and the considerable latent heat release have a noticeable impact on the South America climate by, for instance, maintaining the Bolivian High, which is a key component of the South American monsoon system (Zhou and Lau 1998; Vera et al. 2006).

Clouds and aerosol particles interact in a unique way in the Amazon. Low concentrations of natural aerosols derived from the forest are the major source of natural cloud condensation nuclei (CCN) and ice



nuclei (IN) populations under undisturbed conditions (Pöschl et al., 2010; Prenni et al., 2009; Pöhlker et al., 2012, 2016). Other sources of aerosol particles over the Amazon include long range Saharan dust and sea salt transport, biomass burning (either naturally-occurring or human-induced) and urban pollution downwind from cities and settlements (Talbot et al., 1988,1990; Cecchini et al., 2016; Martin et al., 2010; Kuhn et al., 2010).

Human-emitted pollution can significantly alter cloud properties by enhancing CCN number concentrations ($N_{CCN}$). Since the work of Twomey (1974) analyzing the effects of enhanced $N_{CCN}$ on cloud albedo, much attention has been given to aerosol-cloud-precipitation interactions. The effects of aerosol particles on warm-phase precipitation formation is fairly well understood, where enhanced CCN concentrations lead to the formation of more numerous but smaller droplets delaying the onset of rain (Albrecht 1989; Seifert and Beheng 2006; van den Heever et al. 2006; Rosenfeld et al. 2008). However, in mixed-phase clouds, the rain suppression by pollution can enhance ice formation leading to stronger updrafts and convective invigoration (Andreae et al. 2004; Khain et al. 2005; van den Heever et al. 2006; Fan et al. 2007; van den Heever and Cotton 2007; Lee et al. 2008; Rosenfeld et al. 2008; Koren et al. 2010; Li et al. 2012; Gonçalves et al., 2015). Aerosol effects on clouds have been reviewed by Tao et al. (2012), Rosenfeld et al. (2014), and Fan et al. (2016). By changing cloud properties, aerosol particles have an indirect impact on the thermodynamics of local cloud fields through, for instance, the suppression of cold pools and the enhancement of atmospheric instability (Heiblum et al. 2016b).

Clouds that develop above the freezing level are more difficult to model given the complexity of the processes involving ice particles. One aspect of the aerosol effects on clouds is their ability to alter the way in which ice is formed in the mixed phase of convective clouds. Contact freezing is possibly the dominant process by which the first ice is formed (Cooper, 1974; Young, 1974; Lamb et al., 1981; Hobbs and Rangno, 1985). As pointed out by Lohmann and Hoose (2009), anthropogenic aerosol particles can either enhance or hinder cloud glaciation due to primary aerosol emission (increasing IN concentrations) and aerosol particle coating (decreasing IN effectiveness), respectively. After the initial ice formation, secondary ice generation can be triggered by the release of ice splinters from freezing droplets (Hallett and Mossop 1974; Huang et al. 2008; Sun et al. 2012; Lawson et al. 2015). Rather big (larger than 23 µm) cloud and drizzle droplets favor secondary ice generation (Mossop 1978; Saunders and Hosseini



2001; Heymsfield and Willis 2014). Consequently, the smaller droplets found in polluted Amazonian clouds (Andreae et al. 2004; Cecchini et al. 2016; Wendisch et al. 2016) may slow down secondary ice generation.

In order to model aerosol effects on clouds and the thermodynamic feedback processes involved, it is crucial to understand their effects on hydrometeor size distributions. The first step is the study of aerosol impacts on liquid droplet size distributions (DSDs) in the cloud's warm-phase. Operational models that require fast computations usually adopt a Gamma function (Ulbrich, 1983) to parameterize the DSDs:

$$N(D) = N_0 D^\mu \exp(-\Lambda D) \tag{1}$$

where $N_0$ (cm$^{-3}$ μm$^{-1-\mu}$), $\mu$ (dimensionless), and $\Lambda$ (μm$^{-1}$) are the intercept, shape, and curvature parameters. $N(D)$ is the concentration of droplets per cm$^{-3}$ of air and diameter ($D$) bin interval. Even though this function is widely adopted in models (Khain et al. 2015), there is almost no study regarding its phase space for checking DSD predictions between parameterization schemes.

The phase space of cloud micro- and macro-physical properties has received recent attention because of the considerable gain of information accessible by relatively simple analysis tools. Heiblum et al. (2016a, b) studied cumulus fields in a two-dimensional (2D) phase space consisting of cloud center of gravity versus water mass. The authors were able to evaluate several processes in this sub-space, including the aerosol effect. McFarquhar et al. (2015) studied the Gamma phase space for improving ice particle size distribution (PSD) fitting and parameterization. They showed that the inherent uncertainty of Gamma fittings results in multiple solutions for a single ice PSD, corresponding to ellipsoids rather than points in the phase space. However, there is no study regarding the representation of warm-phase cloud DSDs in the Gamma phase space and its evolution.

For the representation of hydrometeor size distributions in two-moment bulk schemes, one of the three Gamma parameters is either fixed or diagnosed based on thermodynamic or DSD properties (Thompson et al. 2004; Milbrandt and Yau 2005; Formenton et al. 2013a, 2013b). This process may produce artificial trajectories in the phase space when comparing Gamma fittings to observations. This study analyzes cloud DSD data collected during the ACRIDICON-CHUVA campaign (Wendisch et al. 2016) in the Gamma phase space. ACRIDICON is the acronym for "Aerosol, Cloud, Precipitation, and Radiation Interactions and Dynamics of Convective Cloud Systems", while CHUVA stands for "Cloud Processes of the Main



Precipitating Systems in Brazil: A Contribution to Cloud Resolving Modeling and to the GPM (Global Precipitation Measurement)". The Gamma phase space and its potential use for understanding cloud processes is introduced and explored. A specific focus is on the aerosol effect on the trajectories in the warm-layer phase space and potential consequences for the mixed-phase formation.

Section 2 describes the instrumentation and methodology. The results are presented in Section 3, followed by concluding remarks in Section 4.

## 2 Methodology

### 2.1 Flight characterization

During September-October 2014, the German HALO (High Altitude and Long Range Research Aircraft)
performed a total of 96 h of research flights over the Amazon. The 14 flights were part of the ACRIDICON-CHUVA campaign (Machado et al. 2014; Wendisch et al. 2016) that took place in cooperation with the second intensive operation period (IOP2) of the GoAmazon2014/5 experiment (Martin et al. 2016). Here we focus on cloud profiling sections during six flights that occurred in different regions in the Amazon (Figure 1). The research flights of ACRIDION-CHUVA were named chronologically from
AC07 to AC20; the six flight missions focusing on the profiling of cloud microphysical properties (AC07, AC09, AC12, AC13, AC18 and AC19) accumulated 16.8 h of data (in or out of clouds), of which 50 min were inside the lower 6 km of the clouds. We concentrate on these flights for the DSD analysis in order to capture both warm-phase characteristics and early mixed-layer formation. The time frame of the campaign corresponds to the local dry-to-wet season transition, when biomass burning is active in the southern
Amazon (Artaxo et al. 2002, Andreae et al. 2015).

The flight paths followed a regular three-stage pattern: (i) Sampling of the air below clouds for aerosol characterization, (ii) Measurements of DSDs at cloud base, and (iii) Sampling of growing convective cloud tops (Braga et al., 2016; Wendisch et al., 2016). Surface and thermodynamic conditions were different for the various flights (see Figure 1 and 3) with high contrasts in the north-south direction. Logging,
agriculture, and livestock activities management involve burning extended vegetated areas in the region, which emits large quantities of particles that serve as CCN in the atmosphere (Artaxo et al. 2002; Roberts



et al. 2003). Because of this, this region is known as the "Arc of Deforestation," and its thermodynamic properties tend toward pasture-like characteristics. The energy partitioning over pasture-like areas is different compared to regions over the rainforest (Fisch et al. 2004), favoring sensible heat flux and higher cloud base heights (see Table 1).

5 The cloud profiling missions were mostly characterized by cumulus fields, with some developed convection in two flights over the Arc of Deforestation (Figure 2d, f). For flight AC07 some precipitation-sized droplets were observed (not shown); the clouds sampled during AC12 and AC13 presented almost no droplets having $D > 100$ µm. The precipitation during AC07 might be explained by the lower aerosol particle number concentrations compared to flights AC12 and AC13, later start time of the profile, and 10 the presence of deep convection nearby (Table 1 and Figure 2).

## 2.2 Data handling and filtering

The results to be presented here are based on five sensors carried by HALO. A comprehensive description of the airborne instrumentation introduced below can be found in Wendisch et al. (2013). Aerosol particle number concentrations ($N_{CN}$) were measured by a butanol-based Condensation Particle Counter (CPC). 15 The flow rate was set to 0.6 L min⁻¹, with a nominal cut-off particle size of 10 nm. $N_{CCN}$ at a given supersaturation (*S*, averaging 0.48% ± 0.033% for the data used here, with 10% error) was measured by a Cloud Condensation Nuclei Counter (CCN-200, Roberts and Nenes 2005). This instrument contains two columns and was connected to two different inlet systems for aerosol sampling: the HALO Aerosol Sub-micrometer Inlet (HASI) for the aerosol particles and the Counter-flow Virtual Impactor (CVI) inlet 20 to sample cloud droplets, evaporate the cloud water, and analyze the residual particles. The aerosol measurements reported in this study refer to the HASI inlet.

Cloud DSDs were measured by a Cloud Droplet Probe (CDP, Lance et al. 2010; Molleker et al. 2014) that is part of the Cloud Combination Probe (CCP). The CCP also contained a grayscale Cloud Imaging Probe (CIPgs, Korolev 2007), but we focus on CDP measurements where $D < 50$ µm. The CDP counted 25 and sized the droplets based on their forward scattering characteristics, sorting them into 15 droplet size bins between 3 µm and 50 µm. The sample volume had an optical cross-section of 0.278 mm² (±15%). Uncertainties in the cross-section area, the sampling volume, and counting statistics were the major





sources of uncertainty for the DSD measurements (Weigel et al., 2016). We excluded all cloud DSDs with droplet number concentrations ($N_d$) less than 1 cm$^{-3}$ from further analysis.

The DSDs measured by the CDP were fitted to Gamma distributions (Eq. 1) by matching the zeroth, second and third moments. These moments were chosen in order to favor the study of the DSD properties of interest to this study (i.e., droplet number concentration, liquid water content, and effective diameter), but they also coincided with the properties usually predicted by bulk microphysics models (zeroth and third moments in two-moment schemes). The Gamma parameters are calculated by:

$$\mu = \frac{6G - 3 + \sqrt{1 + 8G}}{2(1 - G)} \tag{2}$$

$$\Lambda = \frac{(\mu + 3)M_2}{M_3} \tag{3}$$

$$N_0 = \frac{\Lambda^{\mu+1} M_0}{\Gamma(\mu+1)} \tag{4}$$

where $M_p$ is the *p-th* moment of the DSD. The symbol $G$ is a non-dimensional ratio, given as follows:

$$G = \frac{M_2^3}{M_3^2 M_0} \tag{5}$$

The three parameters $N_0$, $\mu$, and $\Lambda$ define the Gamma distribution in Eq. (1); they are used to construct the phase space described in the next section. The DSD bulk properties, such as droplet number concentration ($N_d$), liquid water content (*LWC*), effective droplet diameter ($D_{eff}$), and relative dispersion ($\varepsilon$), can be derived from the Gamma parameters $N_0$, $\mu$, and $\Lambda$ by taking into account the complete Gamma function integral properties:

$$N_d = \int_0^\infty N(D)dD = N_0 \frac{\Gamma(\mu+1)}{\Lambda^{\mu+1}} \tag{6}$$

$$LWC = 10^{-9} \frac{\pi}{6} \rho_w \int_0^\infty N(D)D^3 dD = 10^{-9} \frac{\pi}{6} \rho_w N_0 \frac{\Gamma(\mu+4)}{\Lambda^{\mu+4}} \tag{7}$$

$$D_{eff} = \frac{\int_0^\infty N(D)D^3 dD}{\int_0^\infty N(D)D^2 dD} = \frac{\mu+3}{\Lambda} \tag{8}$$

$$\varepsilon = \frac{\sigma}{D_g} = \frac{1}{\sqrt{\mu+1}} \tag{9}$$

where $\rho_w = 1000$ g m$^{-3}$ represents the density of liquid water and $\sigma$ and $D_g$ are the DSD standard deviation and mean geometric diameter, respectively. $N_d$, *LWC* and $D_{eff}$ are given in cm$^{-3}$, g m$^{-3}$ and μm, respectively. Given the choice of the conserved moments, they exactly match the respective characteristics of



the observed DSDs. The parameter $\varepsilon$ is described in detail in Tas et al. (2015). The relative dispersion of the Gamma DSD may differ from the observations. Our measurements show that the Gamma and observed $\varepsilon$ are closely related by $\varepsilon_{Gamma} = 0.95\varepsilon_{Observed}$ ($R^2 = 0.93$), showing that the Gamma DSDs are narrower on average. We focus on $\varepsilon$ as obtained by the Gamma parameters and do not use subscripts.

Cloud hydrometeor sphericity was analyzed by the NIXE-CAPS probe (New Ice eXpEriment – Cloud and Aerosol Particle Spectrometer, Luebke et al., 2016, Costa et al., 2017). NIXE-CAPS also contains two instruments, a CIPgs as the CCP and the CAS-Depol for particle measurements in the size range 0.6 to 50 µm. The sizing principle of CAS-Depol is similar to the CDP, the difference is the particle probing: while CAS-Depol has an inlet tube (optimized with respect to shattering), CDP is equipped with an open

path inlet. In addition to the sizing, CAS-Depol is equipped with a detector to discriminate between spherical and aspherical particles by measuring the change of the polarized components of the incident light. Spherical particles do not strongly alter the polarization state, in contrast to non-spherical ice crystals. The cloud particle phase of the whole cloud particle size spectrum was analyzed from the combination of phase determination in the size ranges < 50 µm (from the CAS-Depol polarization signal) and > 50 µm

(from visual inspection of the CIPgs images) (for details, see Costa et al., 2017). Here, the phase states are defined as follows: "Sph (liquid)" stands for many only spherical (D < 50 µm) and predominantly spherical (D > 50 µm) hydrometeors, "Asph small (mixed phase)" for many predominantly spherical (D < 50 µm) and only aspherical (D > 50 µm), "Asph large (ice)" for only very few aspherical (D < 50 µm) and only aspherical (D > 50 µm).

Meteorological conditions, including three-dimensional (3D) winds, were obtained by the Basic HALO Measurement and Sensor System (BAHAMAS) located at the nose of the aircraft (Wendisch et al., 2016). The wind components were calibrated according to Mallaun et al. (2015), with an uncertainty of 0.2 m s⁻¹ and 0.3 m s⁻¹ for the horizontal and vertical directions, respectively. All probes were synchronized with BAHAMAS and operated at a frequency of 1 Hz. All HALO instruments are listed in Wendisch et al.

25 (2016).



## 2.3 Introducing the Gamma phase space

The Gamma fit parameters can be plotted in a 3D subspace where each parameter ($N_0$, $\mu$, and $\Lambda$) represents one dimension. Each point in this 3D Gamma phase space is defined by one ($N_0$, $\mu$, and $\Lambda$) triplet and thus represents one fitted DSD. This space includes all possible combinations of Gamma parameters of the theoretical variability of the DSDs.

The 3D Gamma phase space is illustrated in Figure 3. There are two points in this figure defined by two location vectors $P_1$ and $P_2$, each one representing a fit to a specific DSD (see the insert in the left side of Figure 3) at different times ($t_1$ and $t_2$ for $t_2 > t_1$). If we consider that $\overrightarrow{P_1}$ and $\overrightarrow{P_2}$ represent the same population of droplets evolving in time (i.e., a Lagrangian case), we can link the two points by a displacement vector $\vec{P} = \overrightarrow{P_2} - \overrightarrow{P_1}$, which can be associated to a pseudo-force $\vec{F}$ (blue arrow in Figure 3). We use the term pseudo-force in order to illustrate that the growth processes produce displacements in the phase space. Alternatively, displacements in the phase space can also be understood as phase transitions, in which case each phase is related to a DSD. The pseudo-force $\vec{F}$ can be decomposed into two components, one related to condensational growth and the other to the collision-coalescence (collection) process. The respective pseudo-forces are illustrated as $\overrightarrow{F_{cd}}$ and $\overrightarrow{F_{cl}}$ in Figure 3, respectively. This approach can be applied to multiple points, defining a trajectory through the phase space (gray dotted line). The change of the DSD results in modified Gamma parameters, which determine the trajectory through the Gamma phase space. The direction and speed of the displacements forming the trajectory are determined by the direction and intensity of the underlying physical processes that modify the DSD (condensation and collection). These pseudo-forces are defined by properties such as the initial DSD, CCN, updraft speed, and supersaturation. Of course, this generalization considers only condensation and collision-coalescence. The pseudo-forces can be represented with more sophistication in models, including the several processes involved in DSD changes, such as evaporation, turbulence, melting from the layer above, breakup, sedimentation, etc. Therefore, these two processes can be replaced by a number of pseudo-forces as function of the level of sophistication of the model. We should remember that this approach does not consider contributions from other levels because advection is not directly addressed. To describe the whole process





of DSD evolution during the entire cloud life cycle, the contribution from other layers should be considered.

The direction of the $\overrightarrow{F_{cd}}$ pseudo-force in Figure 3 represents the transition of the DSD during the condensation process, which favors high values of $\mu$ while slightly increasing $\Lambda$. This induces both the narrowing

and a slight increase in the effective droplet diameter (see equations in Section 2.2) of the DSD, which is expected from conventional condensation growth theory. Because of the DSD narrowing, the intercept parameter ($N_0$) is also reduced. Condensational growth may cause a broadening of the DSD in specific situations such as at the cloud base of polluted systems. However, this is an exception and most of the time condensational growth leads to DSD narrowing. The collision-coalescence pseudo-force acts in a

significantly different way in the phase space. From theory and precise numerical simulations that solve the stochastic collection equation, it is known that this process leads to DSD broadening (given the collection of small droplets and breakup of bigger ones) and faster droplet growth in size (compared to condensation). In the Gamma phase space, it should be reflected in lower values of $\Lambda$ and $\mu$, the former decreasing at a faster pace. The intercept parameter $N_0$ should remain relatively constant, given that the

effects of increased mean diameter and DSD broadening balance each other. With $N_0$ almost constant, lower values of $\Lambda$ and $\mu$ result in reduced droplet number concentration, which is consistent with theory (see Figure 7).

In Section 3.2, we show Gamma parameters fitted to real DSD observations. It is not feasible to follow fixed populations of droplets in a Lagrangian way with an aircraft. Therefore, the evolutions we analyze

in the Gamma space are not strictly over time. As a compromise, we use the altitude above cloud base ($H$) of the measurements instead of time evolution, given the conditions of the measurements and our data handling. The cloud profiling missions were planned to capture growing convective elements before reaching their mature state, which is the reason why they usually started at around 12:00 local time. Additionally, we only consider DSD measurements where updraft speed $w > 0$ in order to focus on the

ascending part of the growing clouds.



## 3 Results

### 3.1 Aerosol and thermodynamic conditions in different Amazonian regions

The HALO flights are classified according to the region they covered and the respective aerosol and CCN number concentrations (Table 1). Note the close link between region of the measurements and the aerosol concentrations. From the most pristine clouds at the coast to the most polluted cases in the Arc of Deforestation, there is a ten-fold increase in $N_{CN}$. Remote regions in the Amazon have aerosol particle concentrations slightly higher than over the coast, which is one of the reasons for the term "Green Ocean" used for the unpolluted Amazon regions (Williams et al., 2002). Flights AC07, AC12 and AC13 present flight patterns progressively shifted to the south, which are accompanied by increasing values of $N_{CN}$ and $N_{CCN}$. The farther away the flights take place from the forest, and consequently closer to developed regions, the higher are the pollution levels.

Cloud profiles started at the end of the morning or beginning of the afternoon. The flights were specifically planned for this time period because the convective systems are usually in their developing stages at this time. The freezing level varied between 4500 and 5000 m, while cloud base altitudes were more variable (500 to 2000 m), which resulted from the regional meteorological conditions (Figure 4), and which affects the characteristics of the cloud layers. Clouds in the Arc of Deforestation grow from drier air, given the diminished evapotranspiration rate, and form higher in the atmosphere. As a result, there are thinner warm layers in the polluted clouds, which reduces the time available for droplets to grow by collision-coalescence. Flight AC18 was characterized by a just slightly higher depth of the warm layer compared to the polluted clouds, partly due to the lower altitude of the freezing level. Nevertheless, cleaner clouds can present warm layers 1000 m thicker than clouds affected by pollution.

The vertical profile of the relative humidity (RH) should also be taken into account when comparing clouds formed over different regions. Figure 4b shows that all clouds measured formed in an environment with RH between 60 % and 90 % for their lower 2500 m layer, being higher for forested areas compared to the Arc of Deforestation. For 2500 m and above, there was a significant drying of the atmosphere for flights AC19, AC18 and AC12. It is not clear if the other flights presented similar behavior given the relatively low data coverage for this layer. Regardless, surrounding dry air can significantly enhance the entrainment mixing process (Korolev et al., 2016). As pointed out by Freud et al. (2008), the mixing in





Amazonian convective clouds (and also in other regions – Freud et al., 2011) tends toward the extreme inhomogeneous mixing case, where the effective droplet diameter $D_{eff}$ presents almost no sensitivity to the entrainment. Our result largely corroborates this finding (see Figure 11). It should be pointed out, however, that the recent studies by Korolev et al. (2016), Pinsky et al. (2016a), and Pinsky et al. (2016b) show that homogeneous and inhomogeneous mixing can be indistinguishable depending on meteorological conditions and DSD characteristics when considering the time-dependent characteristics of the entrainment process. Mixing processes may have an impact on the shape of the DSDs measured, thus affecting displacements in the Gamma phase space. The specific type of mixing responsible for it, however, is beyond the scope of this work.

## 3.2 Observed trajectories in the Gamma phase space

In this study, we use the Gamma phase space as a means to study DSD variability. As described in Section 2.3, this space is obtained when the DSD measurements are fitted to Eq. 1, and $N_0$, $\mu$, and $\Lambda$ are used as the dimensions of the 3D subspace. In this space, each point represents one DSD. As the different DSDs were obtained close to the cloud top at the time of the cloud development, the ensemble of positions in the Gamma phase space can be hypothesized as the evolution of the DSDs of a typical cloud through stages of its life cycle. The sequential connection of points (here we use cubic spline fits for illustrating purposes) can be considered as trajectories describing multiple processes responsible for the DSD variability observed. The advantage of using this space is that this variability can be readily observed and compared between different cloud life cycles with different properties. Given the relations between Gamma parameters and DSD properties (Section 2.2), the variability of all cloud microphysical properties can also be inferred from the points in the trajectories. We limited the analysis regarding cloud DSDs and the Gamma phase space to the regions in which $w > 0$ in order to capture the developing parts of the growing convective elements.

Figures 5 to 7 show the Gamma phase space for all profiles considered in this study, grouped by region. The coloring represents the altitude above cloud base ($H$), with the 1 Hz measurements shown as small markers. Bigger markers represent averages at every 200-m vertical interval with available information. Curves (or trajectories) represent cubic spline fits to the averaged points. At first glance, it is possible to





see significant overall differences between the trajectories in the different regions, while internal varia-

tions are much weaker. Aerosol concentrations seem to be a key factor controlling warm-phase properties

in the Amazon, so the internal similarities can be attributed to similar pollution conditions. On the other

hand, differences between the regions stem from the different weights of growth processes. Pristine

clouds, like the ones found over the remote Amazon and the coast of the Atlantic Ocean, are characterized

by faster droplet growth with altitude associated with enhanced collisional growth. In the Gamma space,

this is seen as diagonally-tilted trajectories in Figures 5 and 6, contrasting with the more vertical trajecto-

ries found in polluted clouds (Figure 7).

The differences of the DSD variability in each region highlight the relation of growth processes and tra-

jectories in the Gamma phase space. From the theory described in Section 2.3, it is expected that colli-

sional growth results in diagonal trajectories where the droplets get progressively bigger with DSD broad-

ening. Pristine clouds over the coast and remote Amazon show such tilting (Figures 5-6), indicating that

this process is effective in these systems. The more vertically-oriented trajectories of polluted clouds

(Figure 7) show that there is a different balance between condensational and collisional growth. In terms

of the Gamma phase space characteristics, this can be understood as weaker $\overrightarrow{F_{cl}}$ as a result of smaller

droplets and narrower DSDs. This highlights that the interaction between aerosols and collisional growth

occurs mainly through changes in the initial DSD (i.e., $P_1$ in Figure 3). For each point in the Gamma

phase space the collisional pseudo-forces have different intensities and directions, suggesting that a vector

field can be constructed. This could only be achieved by idealized model experiments, however, where

the updraft speeds can also be prescribed.

Condensational growth can also be illustrated by some points in Figures 6 and 7. Under polluted condi-

tions, this type of growth is expected to be dominant close to cloud base where the droplets are too small

to trigger collision-coalescence. In Figure 7, this is seen in the first 2 or 3 points in the trajectories (dark

blue colors), where the points evolve to higher $\mu$ values with altitude. This results in DSD narrowing and

almost opposite displacement in the Gamma space compared to collisional growth. This trend is shifted

when the altitude is reached where collection processes start to become relevant. Another example of

condensational growth can be seen in Figure 6 at 3000 m. At this point, which is close to the freezing



level, there is a sudden increase in the updrafts (see Tables 2 and 3) and consequently increased conden-sation rates. The rapid increase in condensational growth, with no significant changes in collision-coales-cence, tilts the trajectories to a direction similar to that observed close to cloud base in polluted systems. The displacement is closer to the horizontal direction (i.e., the plane $N_0$ x $\mu$), because droplets are growing

concomitantly by collision-coalescence in the cleaner clouds.

The magnitude of the condensational pseudo-force ($\overrightarrow{F_{cd}}$ in Figure 3) also depends on the initial DSD characteristics ($P_1$). Condensational growth rates are inversely proportional to droplet size, meaning that they get weaker higher in the cloud. The different dependences of $\overrightarrow{F_{cd}}$ and $\overrightarrow{F_{cl}}$ on $P_1$ and their balance throughout the warm-phase life cycle ultimately define the cloud trajectory in the phase space. If they can

be mapped with sufficient resolution, covering different updraft and supersaturation conditions, trajecto-ries may be forecast from a single DSD at cloud base and the evolving thermodynamic conditions. Aero-sols are a key aspect in this regard because they significantly change the cloud-base-DSD in the Gamma space (Figures 5-7) and also affect cloud thermodynamics, impacting condensation rates and conse-quently latent heat release. Note that clouds subject to similar aerosol conditions have similarities in their

trajectories represented by small variability along the trajectories of the respective flights (Figures 6-7).

The $\overrightarrow{F_{cd}}$ and $\overrightarrow{F_{cl}}$ tabulation over the Gamma space can potentially be achieved with the help of Lagrangian large-eddy-simulation bin-microphysics models that precisely solve the condensation and collection equations for varying input DSDs and updraft conditions. Initial DSDs can be obtained from observations and analytical considerations. For instance, Pinsky et al. (2012) show an analytical way to obtain the

maximum supersaturation (which is usually a few meters above cloud base) and the relative droplet con-centration. If $D_{eff}$ behaves adiabatically (Freud et al., 2008; Freud et al., 2011) and is linearly correlated to the mean volumetric diameter (Freud and Rosenfeld, 2012), it is possible to estimate the initial DSD based on Gamma-DSD equations and adiabatic theory given that the aerosol population is known. The advantage of such approach is that all DSD characteristics, most notably its shape, would be realistically

represented and there would be no need for fixing or diagnosing (Thompson et al. 2004; Milbrandt and Yau 2005; Formenton et al. 2013a, 2013b) Gamma parameters for various hydrometeor types – which works for specific applications but may be lacking the physical representation of the processes. This study





focuses on introducing the Gamma phase space and its characteristics, and further work is needed if new parameterizations are to be developed.

### 3.3 Contrasts between clean and polluted trajectories

In this section, we focus on flights AC09 and AC12 in order to study the differences between natural and
human-affected clouds in the Gamma space. Figure 8 shows the trajectories of the clouds measured during these flights, where the points related to the averaged DSDs are numbered and the corresponding properties are shown in Tables 2-3. The numbers start at 1 close to cloud base and grow with altitude ("p" stands for "polluted", while "c" is for "clean"). Also presented in Tables 2 and 3 are the adiabatic fractions which correspond to the ratio between the observed and adiabatic LWC. Some observed DSDs and their corre-
sponding Gamma DSDs are shown in Figure 9, highlighting different growth processes.

It is clear from Figure 8 that clean and polluted clouds cover different regions of the Gamma phase space. Nevertheless, it is possible to see that the trajectories can evolve almost in parallel depending on the dominant growth process. Polluted clouds have wider DSDs at cloud base because of the tail to lower diameters (Figure 9), which brings down the value of $\mu$ (see Eq. 9). Given the lower droplet size (Table
3), condensation is efficient and the trajectories evolve in the overall direction of $\overrightarrow{F_{cd}}$ illustrated in Figure 3. From point 1p to 2p, $N_d$ and *LWC* are approximately doubled. Condensational growth seems to be the dominant growth process in the polluted clouds up to the point 3p, corresponding to a cloud depth of 600 m. A similar layer does not exist in cleaner clouds, where there are enough big droplets to readily activate the collision-coalescence growth. Collisional growth dominates the DSD shape evolution between points
1c and 6c for flight AC09 and between 4p and 7p for AC12. Note that the trajectories are almost parallel in this region. Condensation is still active in this period given the increasing *LWC*, but collision-coalescence have a comparatively bigger impact on the overall DSD shape. Both sections of the trajectories represent 1400-m thick layers, but droplet growth and DSD broadening is more efficient in the cleaner clouds (Figure 9). This explains the pronounced tilting of its trajectory, consistent with a stronger $\overrightarrow{F_{cl}}$
pseudo-force.





Eventually, the trajectories reach a point close to the 0 °C isotherm where the updrafts are enhanced given the continued latent heat release. This *w*-enhanced layer can be several hundred meters thick and culminates in narrower DSDs. This is exemplified between points 7c and 9c and between 8p and 10p. Although droplets are still growing by collision-coalescence, the enhanced updrafts increase condensational growth

sufficiently to produce observable effects on the DSDs. Both trajectories evolve in the condensational growth direction, but with slightly different tilting. The tilting is less pronounced in the cleaner clouds given the stronger $\overrightarrow{F_{cl}}$ component. The way in which the DSDs evolve in this region is important for the mixed-phase initiation, given that both primary and secondary ice generation depend on the characteristics of the liquid droplets. The different properties of the polluted and clean DSDs (see Tables 2 and 3, Figures

8 and 9) indicate that ice formation may follow distinct pathways.

Previous studies suggest that droplets bigger than 23 μm at concentrations higher than 1 cm$^{-3}$ favor secondary ice generation, which was identified as the main mechanism for cloud glaciation (Mossop 1978; Saunders and Hosseini 2001; Heymsfield and Willis 2014; Lawson et al. 2015). In order to visualize these conditions in the Gamma phase space, it is interesting to consider constant $N_d$ surfaces. These surfaces

are defined when $N_d$ is fixed in Eq. 6, resulting in a relation of the form $\Lambda = f(N_0, \mu)$ when inverted. Examples are shown in Figure 10, where $N_d = \{10, 100, 1000\}$ cm$^{-3}$ (axes are rotated for clarity). The surfaces are evidently parallel and are stacked in relatively close proximity (at the scale used here). The trajectories evolve through the surfaces depending on their $N_d$, where polluted clouds tend toward higher droplet concentration (i.e., closer to the red surface). These surfaces can be used to delimit specific regions

of interest. Additionally, further DSD properties can be analyzed along these surfaces. Figure 10 highlights the region of 23 μm < $D_{eff}$ < 50 μm with black lines along the surface of $N_d = 10$ cm$^{-3}$. Regarding cloud DSDs (drizzle droplets are not analyzed here, although they also contribute to ice formation), the region delimited by the black lines for the different surfaces of constant $N_d$ can be interpreted as the most favorable for secondary ice generation, thus indicating a quick glaciation process. Note that the trajectory

of the cleaner clouds enters this region while in the *w*-enhanced layer mentioned previously, which corresponds to the transition to temperatures below 0 °C. Polluted clouds are able to produce high droplet number concentrations, but their smaller droplet size means that they are out of the delimited region. More details about the transition to the mixed phase are given in the next section.





The observation of constant $N_d$ surfaces poses an interesting question for parameterizations. In existing two moment schemes, both $N_d$ and *LWC* are predicted. For each pair of such properties, it is possible to define two surfaces (with constant $N_d$ and *LWC*, respectively) based on Eqs. 6 and 7. These surfaces intersect, defining a curve where both properties are conserved. In this curve, the mean volumetric diam-

eter (proportional to the ratio between *LWC* and $N_d$) is also constant. Based on the limited information provided by the model (only two moments for three Gamma parameters), this curve represents the infinite DSD solutions for the undetermined equation system. A good parameterization scheme should be able to choose one of the DSDs that best fits observations. Given the undetermined equation system, other considerations have to be made.

One parameter that varies along the infinite DSD solution curve is the relative dispersion $\varepsilon$. If $\varepsilon$ is calculated from theoretical considerations or provided by observations, it should be possible to obtain the full Gamma DSD – which is the point in the intersection curve that presents the given $\varepsilon$. The advantage of relying on $\varepsilon$ is that it has low variability between clean and polluted clouds and its average is almost constant with altitude. Tas et al. (2015) studied the relative dispersion parameter in detail, noting that

averaged values for $\varepsilon$ were independent of $N_d$, *LWC,* or height but its variability is significantly lower for the most adiabatic portions of the cloud (notably its updraft core). For precise parameterizations, $\varepsilon$ variability should be taken into account at regions with relatively low $N_d$ and LWC, but averaged values may be considered for the updraft cores. Our observations show that $\varepsilon$ is slightly higher in polluted Amazonian clouds compared to the ones measured over remote regions mainly because of their reduced droplet size

(Tables 2 and 3). This can be considered to produce slight corrections to $\varepsilon$ based on CCN number concentrations.

### 3.4 Observations of the mixed phase formation

The Gamma phase space provides an insightful way to study the formation of the mixed phase by providing the history of the warm phase development as a trajectory. Liquid cloud droplet properties are im-

portant for the glaciation process because they determine the probability of contacting ice nuclei (IN) and





the conditions for secondary generation. As shown in the previous sections, different aerosol and thermo-dynamic conditions alter warm phase characteristics and can thus impact the early formation of ice in the clouds.

Figure 11 shows vertical profiles of $N_d$, LWC, $D_{eff}$, and $\varepsilon$ for clouds subject to background and polluted

conditions (flights AC09 and AC12, respectively). It shows the different microphysical properties (1 Hz) of the clouds associated with the trajectories presented in Figures 8 to 10 ($w > 0$). It shows that droplet concentrations are much higher in polluted clouds, which are not depleted with altitude as much as cleaner clouds (Figure 11a, b). The lower effective diameter for clouds over the Arc of Deforestation may contribute to enhanced evaporation, leading to lower adiabatic fractions. As commented in the previous sec-

tion, $\varepsilon$ shows small variations between the flights and do not change much with altitude.

The properties of the DSDs around the 0 °C level in Figure 11 are a significant feature regarding the mixed phase formation. Note that cleaner clouds have a sudden change in behavior right above the freezing level. At this point, there is a fast decrease in LWC, with higher variability in both $D_{eff}$ and $\varepsilon$. This suggests that ice processes have been triggered, disrupting the smooth evolution observed in the warm

phase. In polluted clouds, this transition takes place at considerably different DSD properties. Averaged $N_d$ reaches values above 1000 cm$^{-3}$ (compared to 50 cm$^{-3}$ in cleaner clouds) with very strong updrafts, bringing LWC closer to adiabaticity. However, no significant variability was observed for $D_{eff}$, suggesting that most of the water is still in condensed state.

In order to further detail the characteristics of the hydrometeors in the transition from warm to mixed

phase, we analyzed the sphericity criteria obtained by the NIXE-CAPS probe (Costa et al., 2017). The methodology developed by Costa et al. (2017) indicates whether each individual 1 Hz measurement contained some aspherical hydrometeors or not. This criterion can be used to indicate whether the hydrometeors are liquid (spherical), mixed (spherical and aspherical), or frozen (aspherical). By combining all measurements for clouds over the remote Amazon (AC09 and AC18) and the Arc of Deforestation (AC07,

AC12 and AC13), we obtained the results shown in Figure 12.

The classifications shown in Figure 12 separate the volumes probed as containing only spherical hydrometeors ("Sph (liquid)") or if there are also aspherical particles too. In that case, the data are further divided into containing both small (D < 50 µm) spherical and large aspherical (D > 50 µm) – "Asph small



(mixed phase)" – or if there are only large (D > 50 µm) aspherical particles – "Asph large (ice)". It is possible to observe that close to cloud base most of the hydrometeors were detected as spherical for both regions, which is expected given that it is the warmest layer of the cloud. However, higher in the clouds the distribution of the classifications become different. The amount of measurements with aspherical particles increases relatively fast for the cleaner clouds, being higher than 90% at the layer around 0 °C. For polluted clouds, on the other hand, almost half of the measurements contained exclusively spherical hydrometeors at this level. Exclusively spherical hydrometeors persisted with a frequency of ~20 % down to temperatures of -15 °C. This is in line with previous studies that found supercooled droplets high into continental convective systems (Rosenfeld and Woodley, 2000; Rosenfeld et al., 2008). Our results show that the persistence of supercooled droplets in Amazonian clouds is more likely under polluted conditions. The characteristics of the cloud warm layer determine the properties of the liquid DSDs close to the 0 °C level and should have a determining role in the glaciation initiation. Our measurements show that clean clouds can produce droplets roughly twice the size of the ones found in polluted systems at this layer, at 95 % lower droplet concentrations (Tables 2 and 3). Bigger droplets are not only more likely to interact with IN and glaciate by immersion or contact freezing, but may also trigger a cascading effect through secondary ice generation (Heymsfield and Willis, 2014; Lawson et al., 2015). This process is able to quickly glaciate the cloud, which fits the results shown in Figure 12. Beyond the DSD bulk properties, the Gamma phase space can also provide more information regarding the kind of DSD that enables or inhibits the glaciation process. In the present study, we have only a few examples to compare warm- and mixed-phase characteristics, but it is clear that it is possible to correlate some regions of the phase space with the characteristics of the ice initiation. Detailed model experiments would greatly enrich this discussion by providing control over the liquid DSD properties and the resulting formation of the mixed layer. More specifically, it would be invaluable to study the impacts of the properties of DSDs at cloud base and at the 0 °C isotherm on the primary and secondary ice production.

## 4 Concluding remarks

Despite being widely adopted in many modeling and remote sensing applications, there is almost no study analyzing the evolution of cloud droplet size distributions in Gamma phase space. Here, we introduce this




visualization, defined by the intercept, shape, and curvature of the Gamma curve, which is parameterized by obtaining the moments of order zero, two, and three. We show that trajectories in the space are related to DSD evolution and are linked to microphysical processes taking place inside the cloud. These processes can be understood as pseudo-forces in the phase space.

Measurements over the Amazon during the ACRIDICON-CHUVA and GoAmazon2014/5 campaigns show that it is possible to relate the direction of the pseudo-forces to different DSD growth processes. Cloud layers with strong updrafts and consequently relatively strong condensational growth showed that this process induces displacements in the direction of high shape and curvature parameters. This tendency is accompanied by DSD narrowing, consistent with condensational growth theory. On the other hand,

collision-coalescence, observable in clean clouds over the Amazon, favors displacements in roughly the opposite direction. Observed displacements in the warm phase may be interpreted as a combination of both pseudo-forces.

The Gamma phase space can also be used as a diagnostic tool for cloud evolution. By studying the displacements in the warm phase, it is possible to determine regions that favor, for instance, cloud glaciation.

Previous studies have identified cloud conditions that favor rapid secondary ice generation, which can be translated into the phase space. We show that clean clouds over the Amazon evolve into the region that favors secondary ice generation because of the enhanced collisional growth. Droplets in polluted clouds take much longer to grow by warm processes and they cross 0 °C long before reaching the region favorable for glaciation. This leads to the persistence of supercooled droplets higher in the clouds, which in-

teract with other ice processes including sublimation to produce big ice particles through the Wegener-Bergeron-Findeisen mechanism. In this regard, the Gamma phase space approach proves to be an interesting tool to analyze the relation between warm microphysics and the evolution of the mixed phase. More studies are encouraged in that direction, especially in modeling scenarios given the difficulties in the prediction of mixed phase processes.

We propose that the Gamma space can be used to both evaluate current parameterization and steer the development of new ones. The results presented here show that different types of clouds have different trajectories through the Gamma phase space. The aerosol effect seems to play a major role in the trajectories of the warm layer. The ability of current parameterizations to reproduce such aspects can be tested




in the phase space, where artificially produced DSDs would be apparent. For new two-moment parame-
terizations, the Gamma space can be used to constrain the DSD from the given droplet concentration and
liquid water content. For each pair of these properties, the possible DSD solutions lie on a curve in the
Gamma space where the main differentiating factor is the distribution relative dispersion. Observations

such as the ones shown here and in previous studies can be used to find the appropriate relative dispersion
value to find the optimal solution. Additionally, precise bin microphysics simulations can be used in order
to produce full condensational and collisional pseudo-force fields in the space. The fields would be de-
pendent on the evolution of properties such as aerosol concentration, updraft speed, and supersaturation
conditions. With such a tabulation, bulk microphysical models would only need to predict the initial DSD

close to cloud base and the rest would be determined by the pseudo-force fields.
This paper shows just an initial view of potential applications of the Gamma space. Future efforts are
encouraged in order to test its efficiency and adequacy. Currently, we are performing bin microphysics
simulations in a column model to compare different closures in bulk schemes. Additionally, we are in the
process of testing the use of the Gamma space in a nowcasting scenario based on dual-polarization radar

retrievals.

*Acknowledgements:* The ACRIDICON-CHUVA campaign was supported by the Max Planck Society
(MPG), the German Science Foundation (DFG Priority Program SPP 1294), the German Aerospace Cen-
ter (DLR), the FAPESP (Sao Paulo Research Foundation) grants 2009/15235-8 and 2013/05014-0), and

a wide range of other institutional partners. It was carried out in collaboration with the USA–Brazilian
atmosphere research project GoAmazon2014/5, including numerous institutional partners. We would like
to thank the Instituto Nacional de Pesquisas da Amazonia (INPA) for local logistic help prior, during and
after the campaign. Thanks also to the Brazilian Space Agency (AEB: Agencia Espacial Brasileira) re-
sponsible for the program of cooperation (CNPq license 00254/2013–9 of the Brazilian National Council

for Scientific and Technological Development). The contribution of Dr. Rosenfeld was supported by pro-
ject BACCHUS European Commission FP7-603445. Micael A. Cecchini was funded by FAPESP grant
2014/08615-7. The entire ACRIDICON-CHUVA project team is gratefully acknowledged for collabora-
tion and support. The data used in this study can be found at http://www.halo.dlr.de/halo-db/.



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

**Figure captions**

**Figure 1:** Profile locations and trajectories of interest to this study. The ACRIDICON-CHUVA research flights were labeled chronologically from AC07 to AC20. The labels in the figure reflect the respective flights where the cloud profiling section took place. The colors represent the different regions: green for remote Amazon, blue for near the Atlantic coast, and red for the Arc of Deforestation (different shades for clarity).

**Figure 2:** GOES-13 visible images for flights (a) AC19, (b) AC09, (c) AC18, (d) AC07, (e) AC12 and (f) AC13. Images are approximately 1 hour after the profile start time.

**Figure 3:** Conceptual drawing of the properties of the Gamma phase space in the warm layer of the clouds. The dotted gray line represents one trajectory through the phase space, representing the DSD evolution. $P_1$ is one DSD that grows by condensation and collision-coalescence to reach $P_2$. The displacement represented by the pseudo-force $\vec{F}$ is decomposed into two components - $\overrightarrow{F_{cd}}$ (condensational pseudo-force) and $\overrightarrow{F_{cl}}$ (collisional pseudo-force). Also shown are the two DSDs representative of points $P_1$ and $P_2$.

**Figure 4:** Average vertical profiles of potential temperature (a) and relative humidity (b) for flights over the Atlantic coast, remote Amazon, and Arc of Deforestation. The markers in the left vertical axis in (a) represent the altitude of the 0 °C isotherm for the different flights. Altitudes are relative to cloud base ($H$, negative values are below clouds). θ and RH are calculated as averages of level flight legs outside clouds.

**Figure 5:** Gamma phase space for flight AC19 over the coastal region. Small markers represent 1 Hz data, while bigger ones are averages for 200 m vertical intervals. The continuous black line represents a





cubic spline fit for the averaged DSDs to illustrate its mean evolution. Altitudes are relative to cloud base ($H$).

**Figure 6:** Similar to Figure 5, but for flights AC09 and AC18 over the remote Amazon.

**Figure 7:** Similar to Figure 5, but for flights AC07, AC12 and AC13 over the Arc of Deforestation.

5 **Figure 8:** Observed trajectories for the clouds measured over the remote Amazon during flight AC09 (continuous line) and over the Arc of Deforestation during flight AC12 (dashed line). The numbers shown close to the observed trajectories start at 1 at cloud base and grow with altitude (the respective markers are colored according do altitude above cloud base, $H$). Their respective properties are presented in Tables 2 and 3.

10 **Figure 9:** Averaged DSDs and their respective Gamma fittings for some points in the trajectories of clouds measured over (a) the remote Amazon (flight AC09) and (b) the Arc of Deforestation (flight AC12).

**Figure 10:** Surfaces of constant $N_d$ as calculated by the inversion of Eq. 6. The trajectories for the clouds measured during flights AC09 (blue) and AC12 (red) are also shown. Note that the axes are rotated for 15 clarity.

**Figure 11:** Vertical profiles of the 1 Hz measurements of $N_d$, $LWC$, $D_{eff}$ and $\varepsilon$ for background clouds over the remote Amazon (a, c, e, g) and polluted clouds over the Arc of Deforestation (b, d, f, h). Updraft speeds are colored in log scale, corresponding to $0.1 \leq w \leq 5$ m s$^{-1}$. Horizontal black lines mark the 0 °C level. Magenta curves in (c) and (d) are the adiabatic water content profiles. $H$ is relative to cloud base 20 altitude.

**Figure 12:** Frequency of occurrence of NIXE-CAPS sphericity classifications for (a) the remote Amazon and (b) the Arc of Deforestation. "Sph (liquid)" stands for many only spherical (D < 50 µm) and predominantly spherical (D > 50 µm) hydrometeors, "Asph small (mixed phase)" for many predominantly spherical (D < 50 µm) and only aspherical (D > 50 µm) hydrometeors, and "Asph large (ice)" for only very 25 few aspherical (D < 50 µm) and only aspherical (D > 50 µm) hydrometeors. Temperatures shown on the x-axis are the center for 6 °C intervals, which corresponds to roughly 1-km-thick layers.



**Table captions**

**Table 1:** General characteristics of the cloud profiling missions of interest to this study: condensation nuclei ($N_{CN}$) and CCN concentrations ($N_{CCN}$, with $S = 0.48\% \pm 0.033\%$), cloud base and 0 °C isotherm altitude ($H_{base}$ and $H_{0°C}$, respectively), start and end time and total number of DSDs collected. The data are limited to the lower 6 km of the clouds. The unit for $N_{CN}$ and $N_{CCN}$ is cm$^{-3}$ and the unit for altitudes is in m. Profile start and end are given in local time.

**Table 2:** Properties of the points highlighted in Figure 8 for flight AC09. $H$ is shown as the average of each of the 200-m vertical bins. The adiabatic fraction is defined as the ratio between the observed and adiabatic $LWC$. Adiabatic values for $N_d$, $LWC$ and $\varepsilon$ are shown below the respective observed quantities.

**Table 3:** Properties of the points highlighted in Figure 8 for flight AC12. $H$ is shown as the average of each of the 200-m vertical bins. The adiabatic fraction is defined as the ratio between the observed and adiabatic $LWC$. Adiabatic values for $N_d$, $LWC$ and $\varepsilon$ are shown below the respective observed quantities.



## Tables

**Table 1:** General characteristics of the cloud profiling missions of interest to this study: condensation nuclei ($N_{CN}$) and CCN concentrations ($N_{CCN}$, with $S = 0.48\% \pm 0.033\%$), cloud base and 0 °C isotherm altitude ($H_{base}$ and $H_{0°C}$, respectively), start and end time and total number of DSDs collected. The data are limited to the lower 6 km of the clouds. The unit for $N_{CN}$ and $N_{CCN}$ is cm$^{-3}$ and the unit for altitudes is in m. Profile start and end are given in local time.

| Region | Flight | $N_{CN}$ (cm$^{-3}$) | $N_{CCN}$ (cm$^{-3}$) | $H_{base}$ (m) | $H_{0°C}$ (m) | Start | End | # DSDs |
|---|---|---|---|---|---|---|---|---|
| Atlantic Coast | AC19 | 465 | 119 | 550 | 4651 | 13:17 | 14:57 | 630 |
| Remote | AC09 | 821 | 372 | 1125 | 4823 | 11:30 | 14:21 | 665 |
| Amazon | AC18 | 744 | 408 | 1650 | 4757 | 12:32 | 14:14 | 397 |
| Arc of Deforestation | AC07 | 2498 | 1579 | 1850 | 4848 | 13:49 | 17:16 | 674 |
| | AC12 | 3057 | 2017 | 2140 | 4938 | 12:55 | 15:16 | 381 |
| | AC13 | 4093 | 2263 | 2135 | 4865 | 12:46 | 15:36 | 204 |



**Table 2:** Properties of the points highlighted in Figure 8 for flight AC09. $H$ is shown as the average of each of the 200-m vertical bins. The adiabatic fraction is defined as the ratio between the observed and adiabatic $LWC$. Adiabatic values for $N_d$, $LWC$ and $\varepsilon$ are shown below the respective observed quantities.

| Point | H (m) | $N_d$ (cm⁻³) | $LWC$ (g m⁻³) | $\varepsilon$ | $D_{eff}$ (μm) | T (°C) | UR (%) | w (m s⁻¹) | Adiabatic fraction |
|---|---|---|---|---|---|---|---|---|---|
| 1c | 100 | 214 | 0.079 | 0.19 | 9.2 | 19.9 | 81 | 0.84 | 0.31 |
| 2c | 300 | 238 | 0.15 | 0.22 | 11.1 | 18.6 | 82 | 0.91 | 0.22 |
| 3c | 500 | 218 | 0.25 | 0.24 | 13.8 | 17.5 | 83 | 1.43 | 0.30 |
| 4c | 700 | 227 | 0.34 | 0.28 | 15.2 | 16.6 | 77 | 1.41 | 0.28 |
| 5c | 1100 | 245 | 0.61 | 0.27 | 18.0 | 13.6 | 85 | 1.13 | 0.31 |
| 6c | 1300 | 284 | 0.79 | 0.29 | 18.9 | 12.0 | 80 | 1.03 | 0.34 |
| 7c | 1700 | 231 | 0.79 | 0.28 | 20.1 | 10.6 | 71 | 1.49 | 0.28 |
| 8c | 2300 | 187 | 1.21 | 0.27 | 24.7 | 7.1 | 78 | 1.66 | 0.34 |
| 9c | 3100 | 233 | 1.95 | 0.22 | 26.4 | 3.5 | 64 | 2.79 | 0.47 |
| 10c | 3900 | 54 | 0.61 | 0.34 | 30.9 | -1.2 | 39 | 1.08 | 0.13 |
| 11c | 4100 | 49 | 0.31 | 0.36 | 25.6 | -1.8 | 61 | 0.31 | 0.065 |
| 12c | 4700 | 36 | 0.26 | 0.47 | 28.6 | -4.8 | 67 | 1.30 | 0.053 |
| 13c | 5300 | 39 | 0.42 | 0.40 | 31.4 | -8.1 | 26 | 2.39 | 0.083 |
| 14c | 5900 | 30 | 0.16 | 0.48 | 26.4 | -11.4 | 33 | 3.27 | 0.032 |



**Table 3:** Properties of the points highlighted in Figure 8 for flight AC12. *H* is shown as the average of each of the 200-m vertical bins. The adiabatic fraction is defined as the ratio between the observed and adiabatic *LWC*. Adiabatic values for $N_d$, *LWC* and $\varepsilon$ are shown below the respective observed quantities.

| Point | H (m) | $N_d$ (cm$^{-3}$) | LWC (g m$^{-3}$) | $\varepsilon$ | $D_{eff}$ (μm) | T (°C) | UR (%) | w (m s$^{-1}$) | Adiabatic fraction |
|-------|-------|-------------------|-------------------|---------------|----------------|--------|--------|----------------|--------------------|
| 1p | 100 | 528 | 0.11 | 0.37 | 8.4 | 16.3 | 72 | 1.17 | 0.59 |
| 2p | 300 | 960 | 0.27 | 0.31 | 8.8 | 15.5 | 64 | 1.02 | 0.72 |
| 3p | 500 | 634 | 0.21 | 0.28 | 9.2 | 14.7 | 58 | 1.28 | 0.29 |
| 4p | 700 | 597 | 0.29 | 0.27 | 10.4 | 12.4 | 59 | 0.57 | 0.24 |
| 5p | 1300 | 543 | 0.34 | 0.29 | 11.5 | 6.9 | 65 | 1.13 | 0.15 |
| 6p | 1900 | 1066 | 1.12 | 0.29 | 13.7 | 2.6 | 69 | 0.74 | 0.38 |
| 7p | 2100 | 874 | 0.75 | 0.31 | 12.8 | 2.4 | 62 | 2.89 | 0.26 |
| 8p | 2700 | 477 | 0.62 | 0.32 | 14.8 | 0.4 | 8 | 1.62 | 0.17 |
| 9p | 2900 | 1271 | 1.95 | 0.32 | 15.7 | 0.2 | 5 | 9.36 | 0.52 |
| 10p | 3300 | 1024 | 1.78 | 0.24 | 15.7 | -1.5 | 3 | 5.68 | 0.44 |
| 11p | 3700 | 137 | 0.25 | 0.24 | 16.0 | -3.6 | 4 | 0.26 | 0.06 |





## Figures

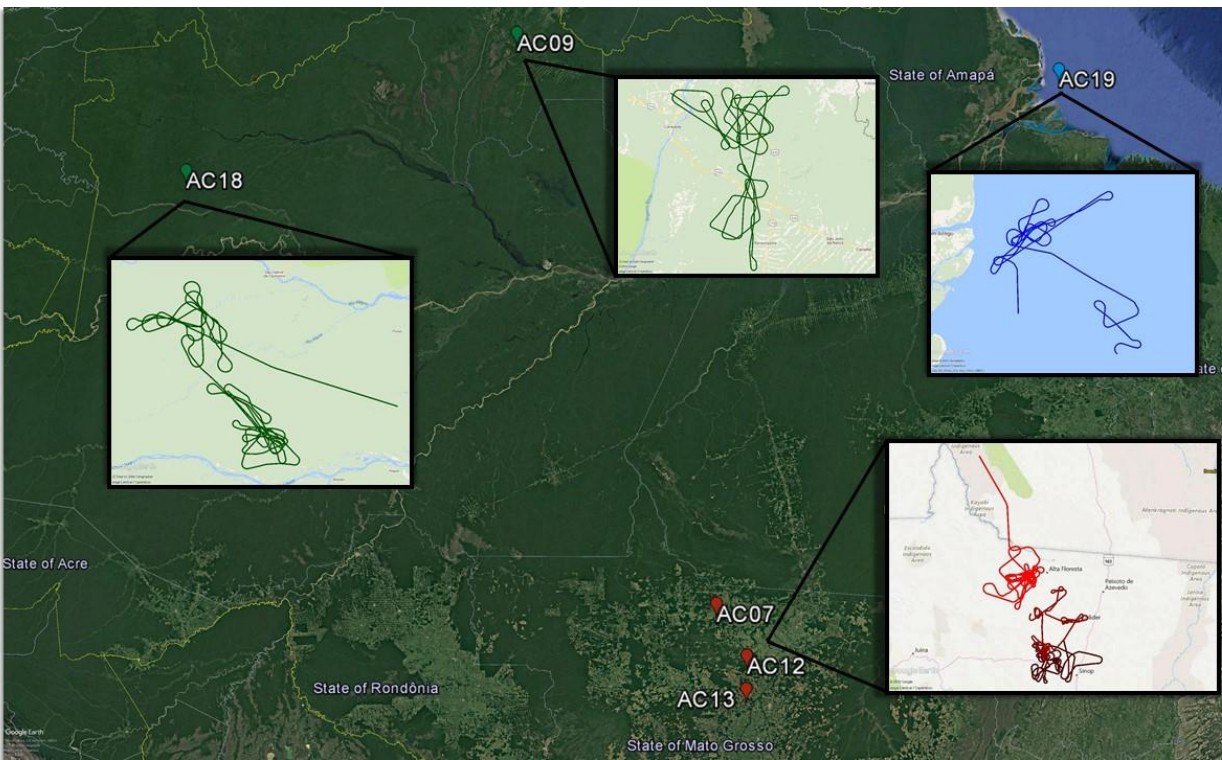

**Figure 1:** Profile locations and trajectories of interest to this study. The ACRIDICON-CHUVA research flights were labeled chronologically from AC07 to AC20. The labels in the figure reflect the respective flights where the cloud profiling section took place. The colors represent the different regions: green for remote Amazon, blue for near the Atlantic coast, and red for the Arc of Deforestation (different shades for clarity).



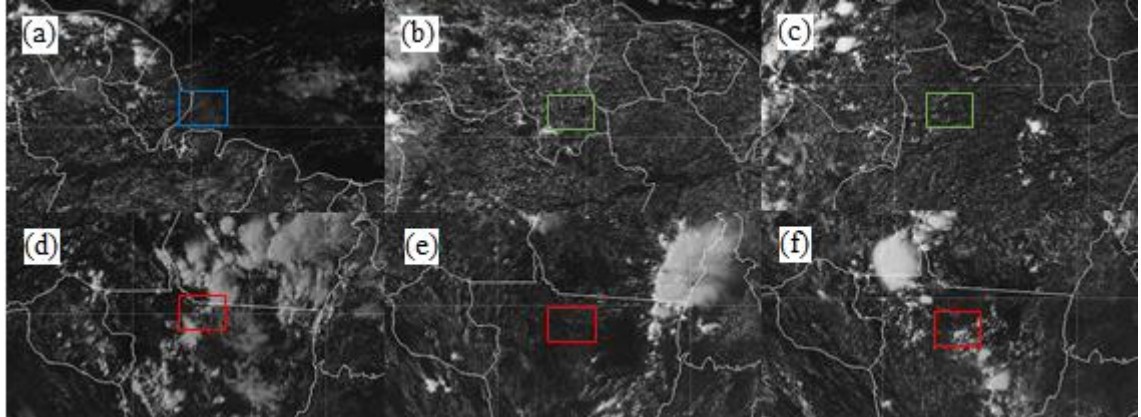

**Figure 2:** GOES-13 visible images for flights (a) AC19, (b) AC09, (c) AC18, (d) AC07, (e) AC12 and (f) AC13. Images are approximately 1 hour after the profile start time.



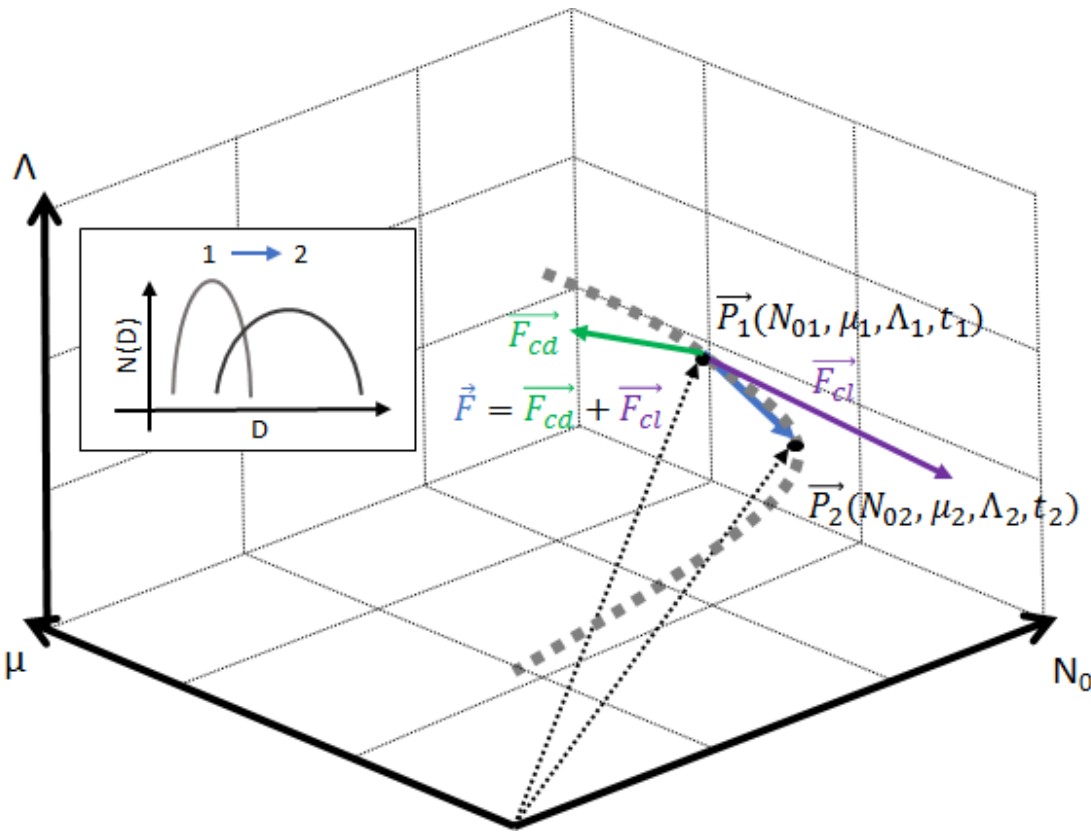

**Figure 3:** Conceptual drawing of the properties of the Gamma phase space in the warm layer of the clouds. The dotted gray line represents one trajectory through the phase space, representing the DSD evolution. $P_1$ is one DSD that grows by condensation and collision-coalescence to reach $P_2$. The displacement represented by the pseudo-force $\vec{F}$ is decomposed into two components - $\overrightarrow{F_{cd}}$ (condensational pseudo-force) and $\overrightarrow{F_{cl}}$ (collisional pseudo-force). Also shown are the two DSDs representative of points $P_1$ and $P_2$.

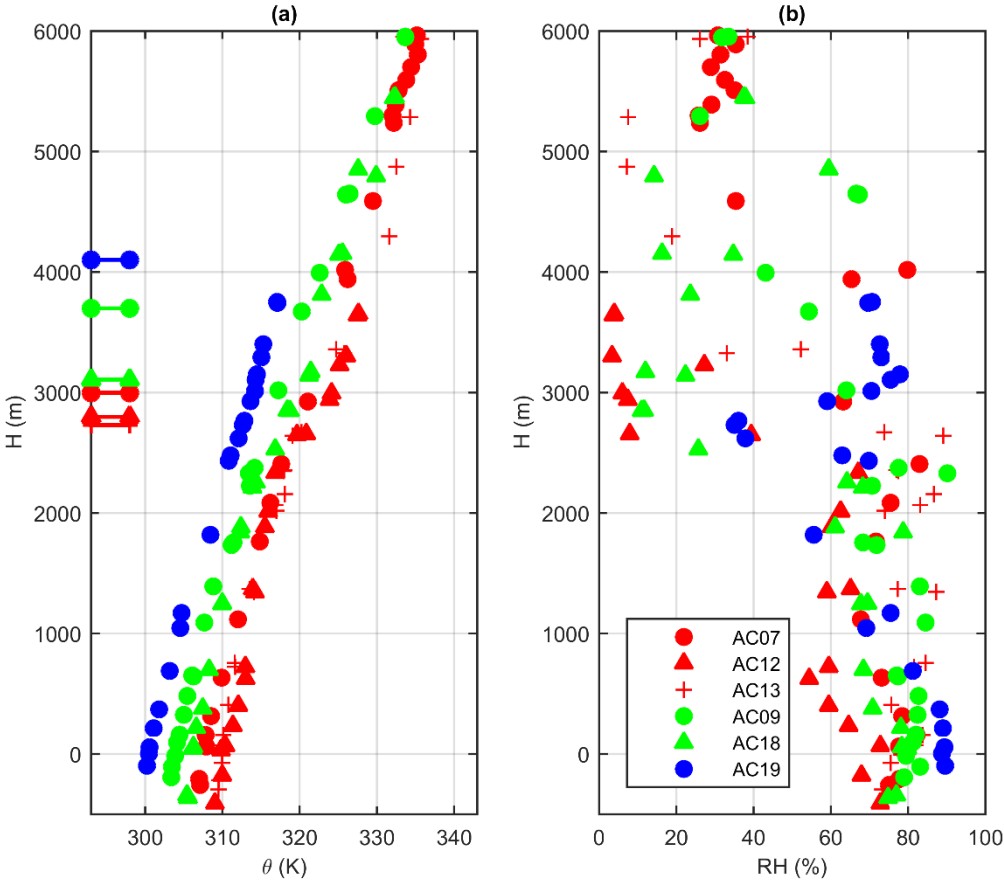

**Figure 4:** Average vertical profiles of potential temperature (a) and relative humidity (b) for flights over the Atlantic coast, remote Amazon, and Arc of Deforestation. The markers in the left vertical axis in (a) represent the altitude of the 0 °C isotherm for the different flights. Altitudes are relative to cloud base (*H*, negative values are below clouds). θ and RH are calculated as averages of level flight legs outside clouds.





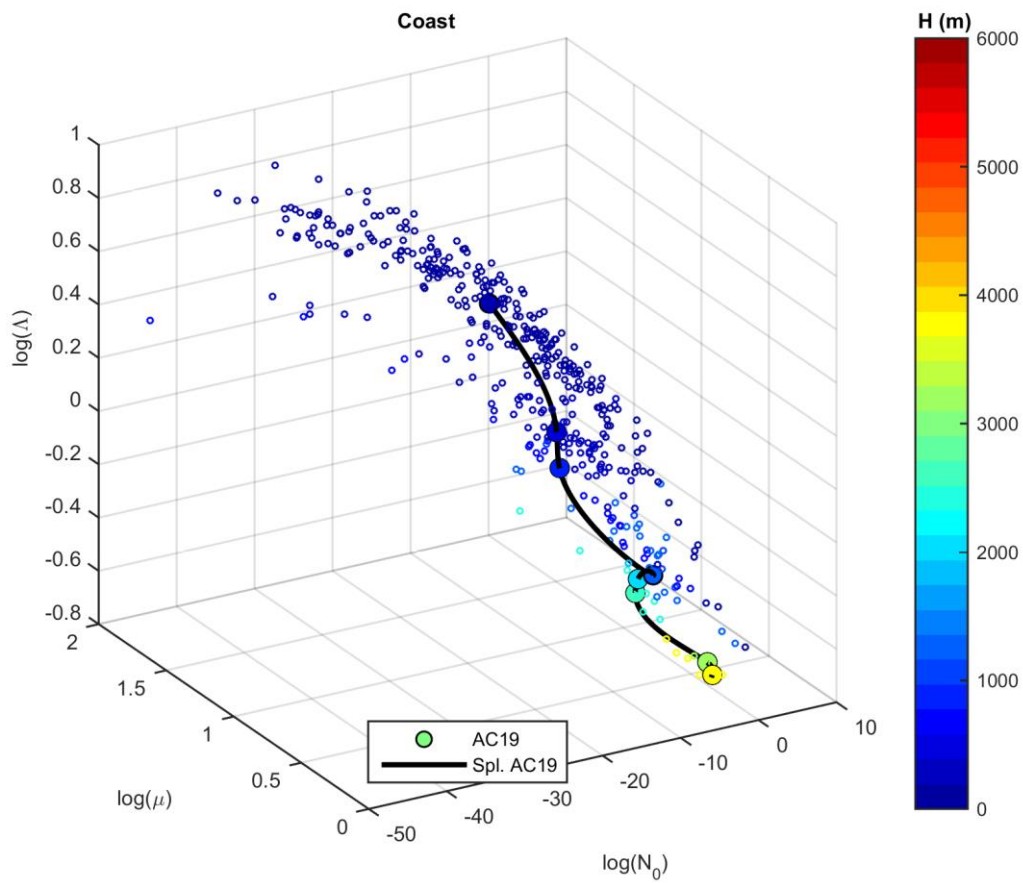

**Figure 5:** Gamma phase space for flight AC19 over the coastal region. Small markers represent 1 Hz data, while bigger ones are averages for 200 m vertical intervals. The continuous black line represents a cubic spline fit for the averaged DSDs to illustrate its mean evolution. Altitudes are relative to cloud base (*H*).

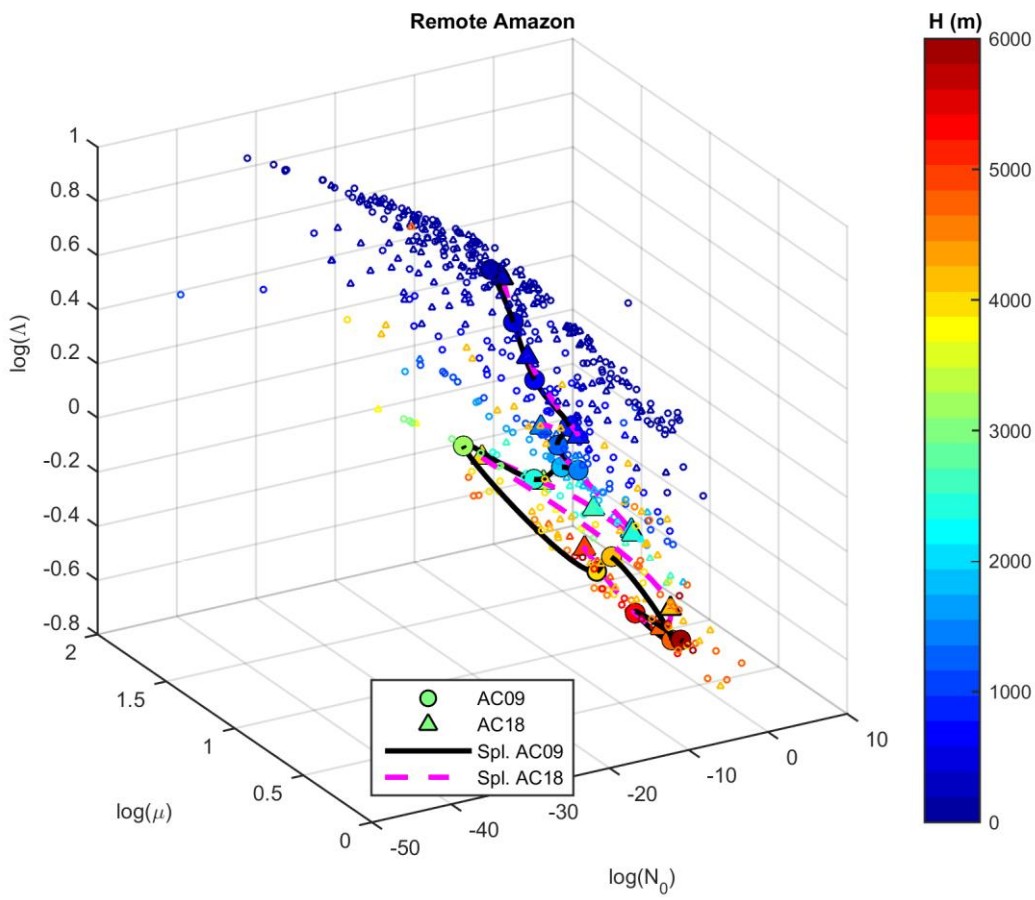

**Figure 6:** Similar to Figure 5, but for flights AC09 and AC18 over the remote Amazon.



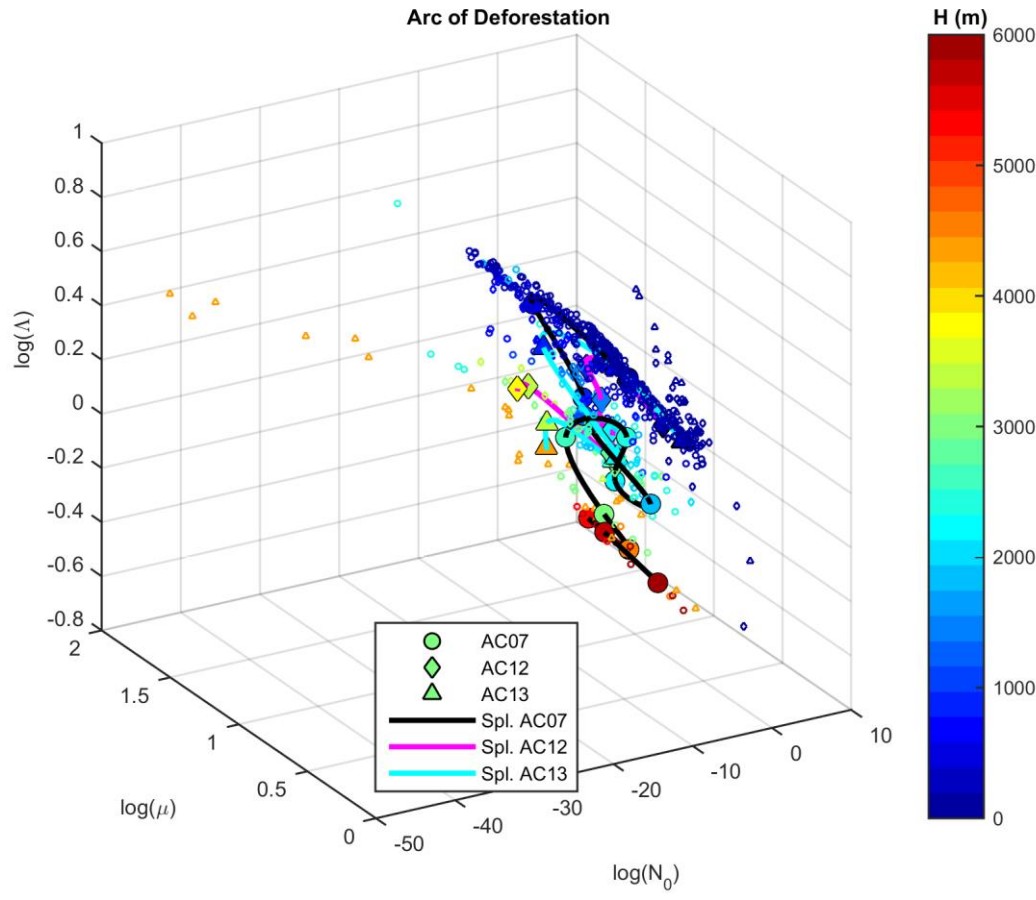

**Figure 7:** Similar to Figure 5, but for flights AC07, AC12 and AC13 over the Arc of Deforestation.




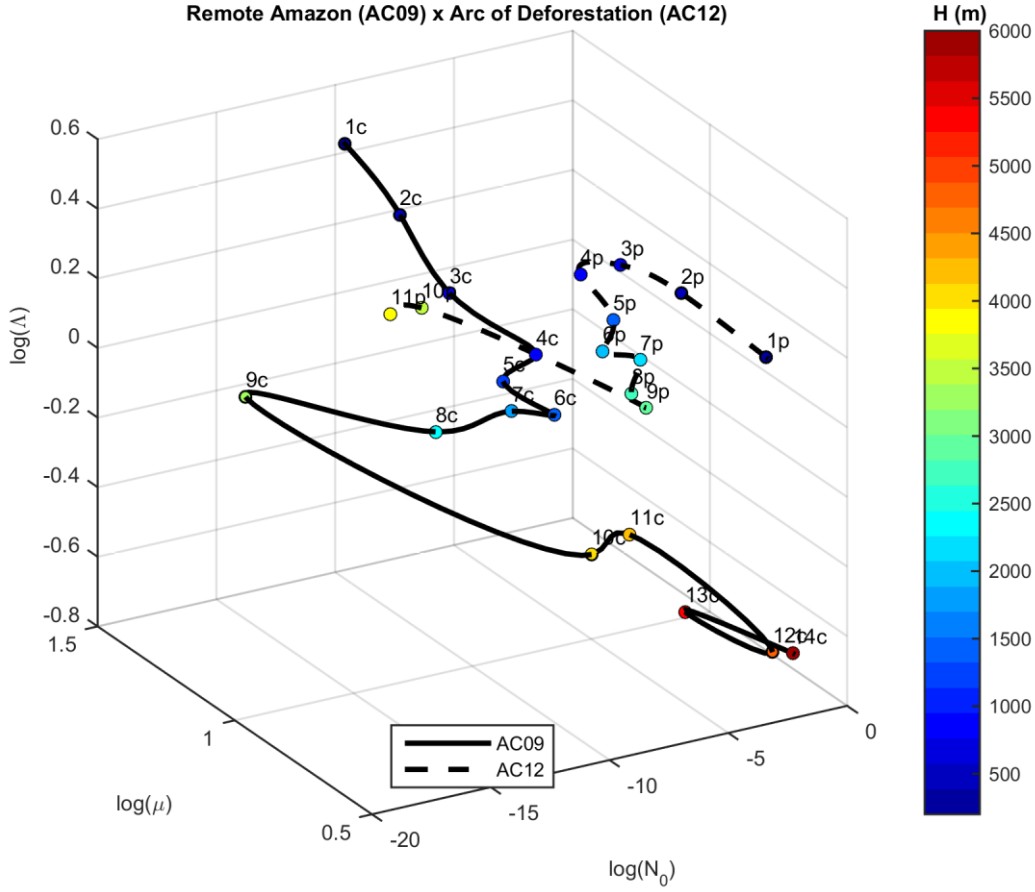

**Figure 8:** Observed trajectories for the clouds measured over the remote Amazon during flight AC09 (continuous line) and over the Arc of Deforestation during flight AC12 (dashed line). The numbers shown close to the observed trajectories start at 1 at cloud base and grow with altitude (the respective markers are colored according do altitude above cloud base, $H$). Their respective properties are presented in Tables 2 and 3.




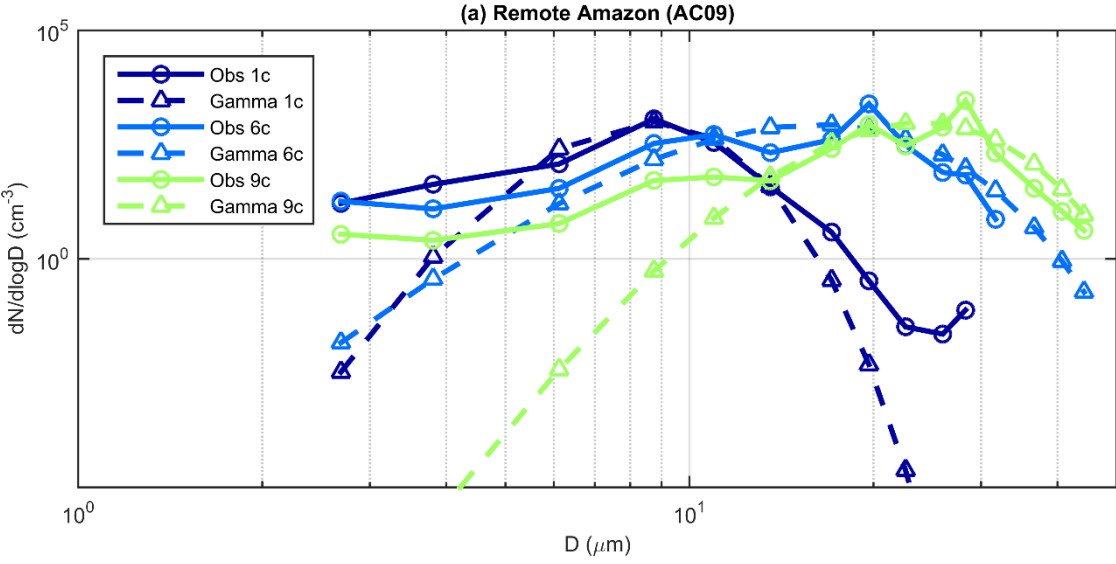

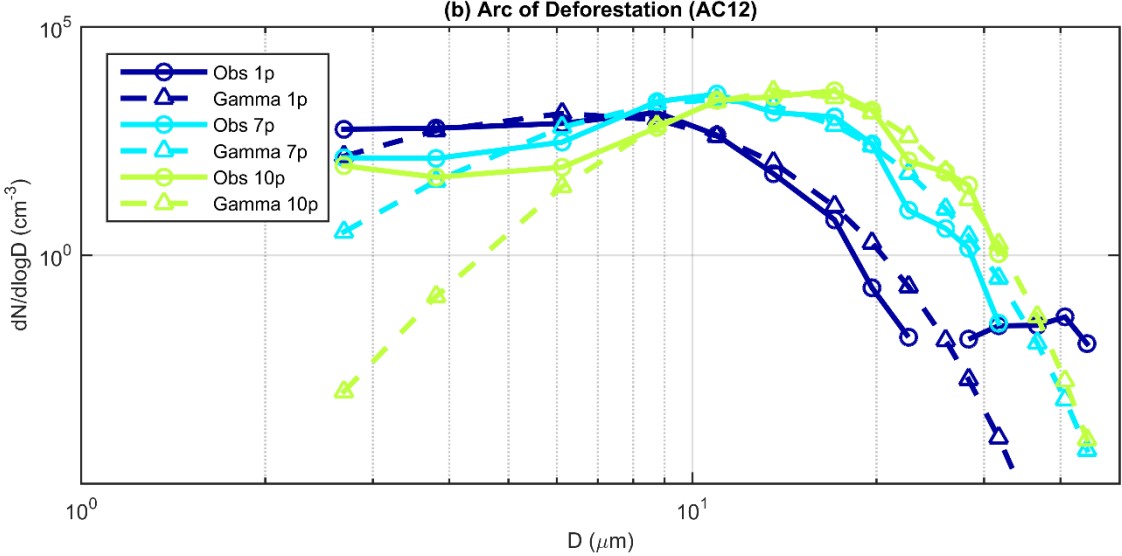

**Figure 9:** Averaged DSDs and their respective Gamma fittings for some points in the trajectories of clouds measured over (a) the remote Amazon (flight AC09) and (b) the Arc of Deforestation (flight AC12).



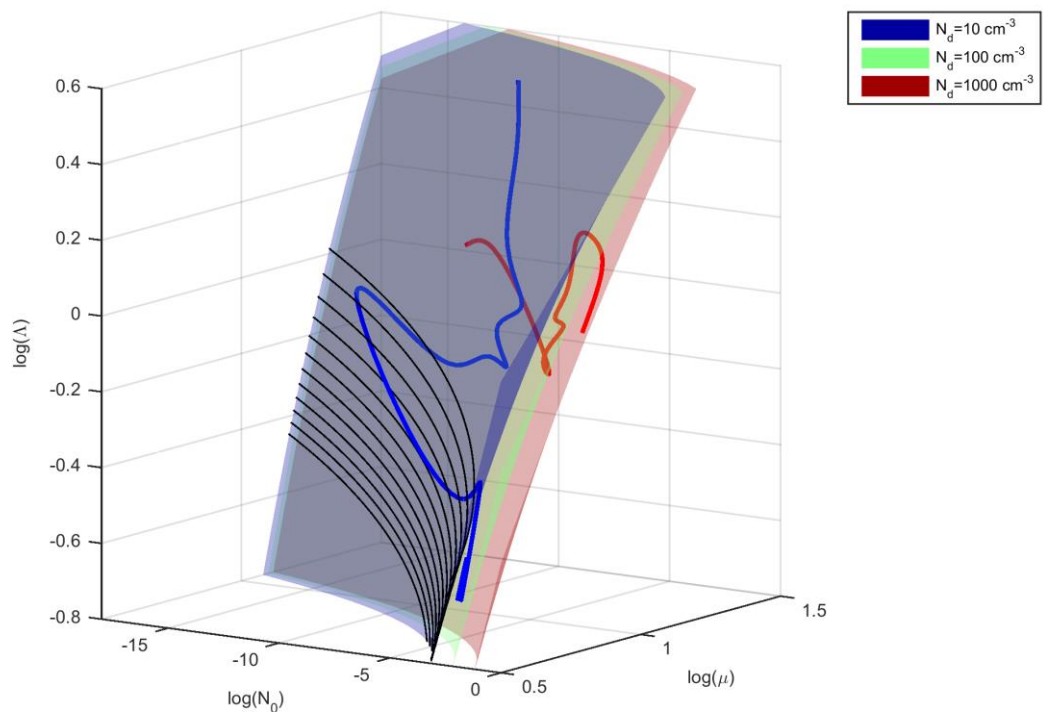

**Figure 10:** Surfaces of constant $N_d$ as calculated by the inversion of Eq. 6. The trajectories for the clouds measured during flights AC09 (blue) and AC12 (red) are also shown. Note that the axes are rotated for clarity.





**Figure 11:** Vertical profiles of the 1 Hz measurements of $N_d$, $LWC$, $D_{eff}$ and $\epsilon$ for background clouds over the remote Amazon (a, c, e, g) and polluted clouds over the Arc of Deforestation (b, d, f, h). Updraft speeds are colored in log scale, corresponding to $0.1 \le w \le 5$ m s$^{-1}$. Horizontal black lines mark the 0 °C level. Magenta curves in (c) and (d) are the adiabatic water content profiles. $H$ is relative to cloud base altitude.





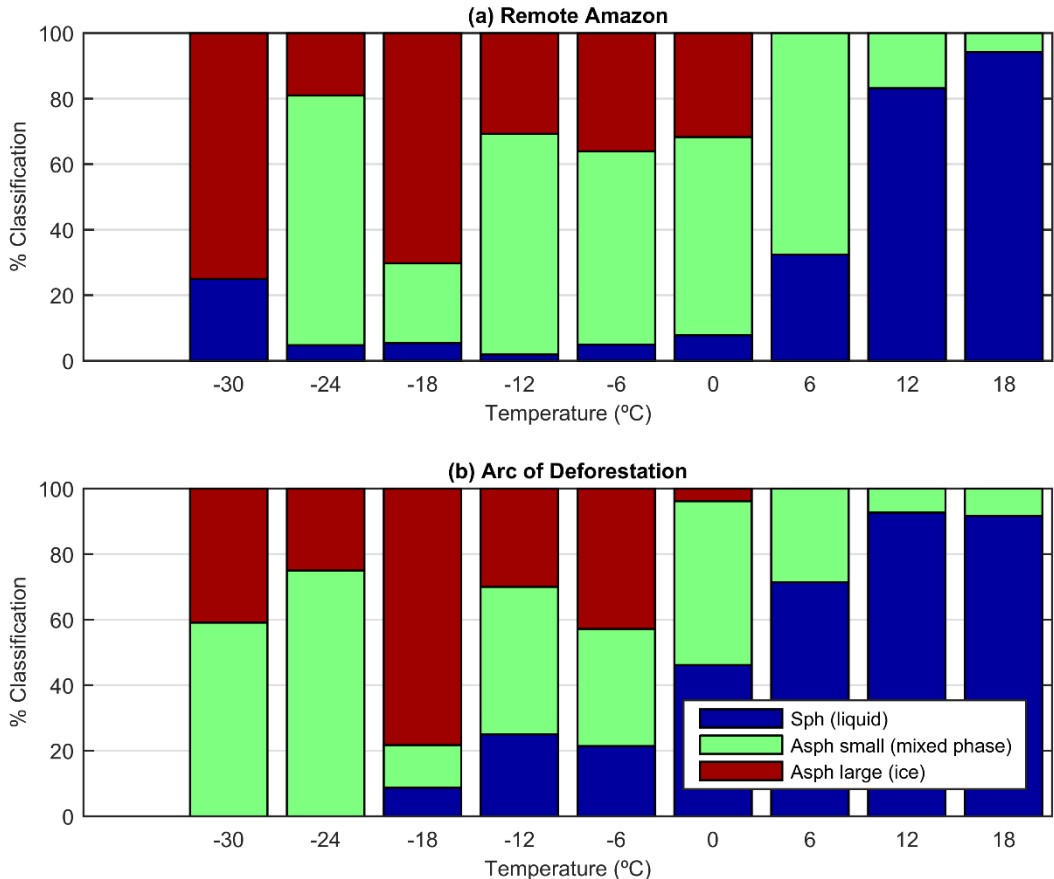

**Figure 12:** Frequency of occurrence of NIXE-CAPS sphericity classifications for (a) the remote Amazon and (b) the Arc of Deforestation. "Sph (liquid)" stands for many only spherical (D < 50 µm) and predominantly spherical (D > 50 µm) hydrometeors, "Asph small (mixed phase)" for many predominantly spherical (D < 50 µm) and only aspherical (D > 50 µm) hydrometeors, and "Asph large (ice)" for only very few aspherical (D < 50 µm) and only aspherical (D > 50 µm) hydrometeors. Temperatures shown on the x-axis are the center for 6 °C intervals, which corresponds to roughly 1-km-thick layers.