# Peer review of "Illustration of microphysical processes in Amazonian deep convective clouds in the Gamma phase space: Introduction and potential applications"

_Atmospheric Chemistry and Physics, 2017_

## Referee Comment (RC1) · Anonymous Referee #1 · 8 Jul 2017

This paper describes an attempt by the authors to use gamma function fitted using cloud macro- and microphysical observations to analyze microphysical growing path and particle size distribution evolution within deep convective clouds. The data used in this study are from ACRIDICON-CHUVA field campaign over Amazon, primarily six flights focusing on cloud microphysics measurements over regions with different aerosol background profiles. The findings of cloud properties under different environments appear to be very interesting. However, the method in using gamma function to interpret cloud microphysical growing path contains serious issues.

[Figure]

The description of the closure of gamma function was firstly given by Eq. (2) - (4), where the closure variables were the zeroth, second, and third moment. The three undetermined parameters of gamma function would be defined by these moments. However, the actual closure variables, as described in Eq. (6) – (8) are liquid water content, cloud droplet number concentration, and effective droplet diameter. The former two were equivalent to the third and zeroth moment, respectively, while the effective droplet diameter was mostly equivalent to the first moment. To the least, these two descriptions are redundant. In fact, in many places of the paper including the Concluding remarks, the authors were still referred to the second moment. Indeed, the procedure of fitting gamma function with observations was never clearly described.

The most serious flaw of the proposed method exists in the procedure to interpret microphysics in the phase space of size distribution function. For a given air parcel, the ternary group of closure variables (mostly moments in different order) and undetermined parameters are bonded by mass conservation applied to the prognostic procedure of the former group, this defines the unique solution of both groups through the evolution of the air parcel, and they change accordingly due to the variations of the closure variables induced by dynamical and microphysical processes. Note also that the closure variables must be conservative ones with well-defined sink and source besides advection and mixing terms. When fitting gamma function with multiple observations, however, one should realize that these observations are multiple snapshots likely represent different air mass origin either unmixed or mixed, therefore, they mostly reflect different ternary groups of the closure variables and hence their paths in the phase space are irrelevant microphysically speaking unless a strong isentropic assumption (at least for any given horizontal plane) is made. This is why even in analyzing Eulerian modeling results, modelers usually derive microphysical and size distribution evolution within a parcel framework (can be conveniently derived from Eulerian grid parameters though), e.g., the "Twomey model". Only within such a framework does the analysis of size distribution evolution become meaningful.

Interactive
comment
By the way, many comments made by the authors are not accurate. For example, in the Abstract, the opening statement seems attempting to link our lack of understanding of the "tropical clouds" solely to the model representation issue of certain physical processes. The statement of "there is almost no study dedicated to understanding the phase space of this function..." is not accurate too. The properties of Gamma function along with many other probability distributions have been well studied and documented in statistics and applied mathematics literature. In the cloud physics and modeling field, the evolution of conservative moments (mostly in the format of LWC, number concentration, and spectral disperse) have never been a rare topic in various mostly modeling studies.

The observations are invaluable for further our understanding of cloud physics and for evaluating models. Applying derivatives of these data, however, warrens special cautiousness.
* * *

---

## Referee Comment (RC2) · Anonymous Referee #2 · 5 Aug 2017

General Comments:

This manuscript introduces a technique for describing cloud processes using the drop size distribution gamma fit coefficients, and the trajectory of these coefficients in three-dimensional space. Comparisons within this phase space are made among clouds with different environmental conditions and linked to various cloud processes. While the manuscript is well written, I think some aspects of the paper need further work.

First, the physical insights that are provided are not closely linked to the coefficients

themselves, and instead are reworked into pseudo-forces related to condensation and collision processes. However, the method used to decompose the trajectories into these pseudo forces is not clearly described, and as a result I find it difficult to accept many of the explanations behind the patterns in the data.

Secondly, gamma functions often provide good mathematical fits to drop size distributions, but attempting to understand cloud processes using the fit coefficients is fraught with difficulty, which I don't think is addressed sufficiently in this manuscript. Gamma function coefficients can vary substantially depending on the fit method used, the size range over which the fit is made, and the suitability of the underlying size distribution shape to be fit with a gamma. Many of these issues were addressed in the recent publication by McFarquhar et al. (JAS 2014). Using different fitting methods they found that the N0 coefficient, for example, can vary by many orders of magnitude, even when the same moments (1, 2, and 6) are used to make the fit. Using a different set of moments, like the 0th, 2nd, and 3rd used in this manuscript would likely result in even larger changes. Furthermore, the coefficients N0 and mu are inextricably linked, with N0 having the units of $m^{(-4-mu)}$. So as mu changes, N0 will respond mathematically, even though such a change may not represent a physical process.

A more effective method may be to plot the moments themselves in 3D space rather than first fitting them to a gamma function. The moments are more easily linked to known microphysical processes, and if they are computed directly from the distributions do not suffer from the complications of poor fitting. The moments can always be computed from modeled DSDs as well, which would the avoid the further complications introduced when models use restricted gamma parameter spaces. At the very least, I think the authors should investigate the sensitivity of the observed phase-space patterns to different gamma fitting methods, and more clearly identify the source and interpretation of the F_cd and F_cl pseudo-forces.

Specific comments:

[Figure]

Section 2.2: How were DSD shapes that are not well fit by a gamma function handled in the analysis (e.g. bimodal or skewed distributions)?

Section 2.3: The introduction of the F_cd and F_cl pseudo-forces seem incomplete and leaves many unanswered question, such as: How were they determined, i.e. can they be presented mathematically? Do they completely describe the total force F? Are they orthogonal, if not, in which direction in the phase space does each force point?

Figures 3,5,6,7, and 8: It is difficult to determine where the lines and points are in 3-D space. A projection of the fit lines onto the X, Y, and Z planes would greatly help with the visualization.

Section 2.3: Given the sensitivity of N0 to the mu parameter, the speculations regarding N_0 would be much more convincing if N_d (or 0th moment) were used instead.

Section 3.2: The manuscript states that measurements were taken 'close to cloud top', but more information is needed about the placement of the measurements in the cloud. Was the aircraft making multiple passes to a fixed location, or attempting to intercept the same visual position in the cloud on mulitple passes? How long did the aircraft pattern take relative to the lifetime of the cloud, and at what point the life cycle of the cloud were the measurements taken?

Section 3.3: How were the clean and polluted clouds determined? Were the flight patterns and environmental conditions for each of these clouds comparable?

---

## Referee Comment (RC3) · Anonymous Referee #3 · 9 Aug 2017

Review of "Illustration of microphysical processes in Amazonian deep convective clouds in the Gamma phase space: Introduction and potential applications" by Cecchini et al.

Recommendation: Requires revision before publication

This paper uses fits of measured cloud droplet size distributions (DSDs) in gamma phase space to investigate warm-phase microphysical cloud properties and the role of "pseudo-forces" in affecting the evolution of the gamma parameters and the DSDs.

[Figure]

Overall, I found the description of a unique set of data interesting and formative, and hence believe that the paper is worthy of publication. However, there are some issues that should be addressed in order to improve the presentation before the paper is published as discussed below.

Major Comments

1. The authors segregate the flights that are flown into the different regions of the Amazon where they are flown. Although changes in surface conditions are no doubt important for affecting the cloud properties, meteorological conditions can also have an important impact on cloud properties. Some comments about this should be added to the manuscript and some analysis of the meteorology on the different days should be added to see if such differences can also explain some of the variation in cloud properties. I think attributing much of the changes to aerosols is not fully justified until the meteorological context is further explored.

2. I was a bit surprised on page 6 where the authors described that they were focusing on the CDP measurements where D < 50 micrometers. It would seem to me to be quite important to also examine the drizzle sized drops measured by the CIP, as when drizzle was present it would seem to be very important to account for that in the analysis. How are flights handled when there was some precipitation-sized drops? Were these larger drops incorporated into the analysis of simply ignored? Further, for Eq. (2) to Eq. (4) should the incomplete gamma function rather than the gamma function be used to account for the fact that not the complete size range of particles were measured?

3. I think some more comments on the quality of the microphysical measurements are needed. How did the CAS and CDP probes compare? What are the estimated uncertainties in the size distributions? How did the LWC integrated from the CAS or CDP compare with bulk measurements from a hot-wire probe, which I am assuming were made. I am assuming that fits were only done to the liquid distributions, or do you use all the distributions? This should be clearly stated when discussing the phase

partitioning at the bottom of page 8.

4. The implicit basis of the analysis presented in the Gamma phase space is that one is dealing with a Lagrangian case. But, inevitably, with any sort of microphysical measurements different samples of particle populations are being sampled. Further, there can be mixing and dynamic motions in clouds that would affect how the DSDs vary in the gamma phase space. Is there any way of representing these mixing processes on the diagram? I also think the action of the pseudo-forces and the impact of condensational growth and collision-coalescence could be better illustrated on the diagram. Can you show an example size distribution (it can be a theoretical rather than observed distribution if it is easier) and show how the size distribution would change due to simple model calculations of either condensational or collision-coalescence growth. Then, illustrate the location of all 3 DSDs (original, one undergoing condensational growth, and one undergoing collision-coalescence growth) on the gamma phase space and it will be easier for the reader to appreciate how these forces are represented on the diagram. Such simple theoretical/modeling calculations may also help you assess how the DSD characteristics are being affect by homogeneous/inhomogeneous mixing (discussion at top of page 12).

5. I'm wondering if some different terminology could be used to refer to the different flights. Although referring to flight numbers (e.g., AC19, AC18, AC12, etc.) might be very informative for people who were involved in the field program, I continually had to refer back to the table to remember the regions in which the flights were conducted to help interpret the data. Can you refer to them as maybe AC1 (AC19 for Atlantic coast 1), RA1 and RA2 (AC09 and AC18) for remote Amazon, and AD1, AD2 and AD3 (for AC07, AC12, and AC13) so that it is more easy to remember the flights going through the manuscript. Or, maybe some other terminology would also work.

6. With regards to the depiction of the DSDs in phase space, I would find it much easier is some 2-d cross sections were presented in addition to the 3-d volumes (it was hard to follow some of the discussion on the contrasts between clean and polluted trajectories).

It is very hard to visualize how the different parameters are changing on these 3-d plots, so some 2-d cross sections would also offer some supplementary information. Further, what are the uncertainties or range of possible values in the gamma parameters.

Minor Comments:

Page 2, Line 20: I was surprised to see that the undisturbed portions of the rainforest are said to have homogeneous surface properties: compared to oceanic surfaces surely the nature of the forest is somewhat inhomogeneous? On page 5 (lines 20-25), the authors talk about differences in surface and thermodynamic conditions on more of the disturbed areas of the Amazon, so I found that this comment was a bit misleading.

Page 3, Line 1: Typically the term ice nucleating particles (INPs) rather than ice nuclei (IN) now. See Vali (2015).

Page 4, line 9: Unless specific numeric values are quoted, the parameters of the gamma function (or any parameter in general) do not have units associated with them. They could be given in any unit with an appropriate conversion being made. Recommend removing the units in parenthesis.

Page 5, line 23: if the convective clouds were growing, how could you ensure that the third stage was always flown through the growing tops? It would seem that different altitudes below cloud top might have been sampled for the different population of clouds.

Page 7, Eq. (7). I don't think this equation is correct (the factor of 10^-9). Any equation must be unit-independent. Constants for conversions between specific unit sets hence don't belong in equations as those factors will automatically appear when converting between the different units of the variables.

Page 8, line 4: If the fact epsilon obtained by the gamma parameters does not match those from the DSDs suggest that the gamma distribution does not give a good fit to the DSD?

Page 9, line 12: Can you use a different word rather than "phase transitions?" there is some confuse about whether you are talking about phase space or the phase (liquid, mixed or ice) of the cloud particles.

Page 11, line 16: "from drier air", can you list the humidities in Table 1?

Page 11, line 23: Do you mean average RH? Clouds do not form in an environment where the relative humidity at their location is between 60% and 90%. Can you also give some indication of the thickness of the different cloud layers?

Page 12, lines 13-14: How do you know the observations were obtained close to cloud top? Unless you have remote sensing data or some ascents out of cloud, is it conceivable the particular cloud you were sampling extended to a higher height?

Page 13, line 1: What is classified as a significant difference? Was some sort of statistical test applied?

Page 14, line 12: What statistical test was applied to know that the res

---

## Referee Comment (RC4) · Anonymous Referee #4 · 10 Aug 2017

This paper used in-situ data from six flights collected during the ACRIDICON-CHUVA field campaign to explore the linkage between gamma distribution parameter phase space and underlying microphysical processes. Three different environmental conditions, the Atlantic Coast, the remote Amazon, and the Arc of Deforestation were studied, and the differences in the underlying microphysical processes among these regions were compared. The paper fits into the scope of ACP and is generally well written, however, the approach used in this study has severe scientific flaws. Therefore, this paper needs to be revised considerably before it can be published in ACP.

[Figure]

Major comments: 1. Page 5, Line 15-18. Are there only six flights during the whole field campaign focusing on clouds? If not, why other flights are not used? Especially for Atlantic Coast, there is just one flight used. 2. Page 6, Line 22-24: Why PSDs from CIPgs is not used? Only using CDP to create PSDs with D<50um will miss out lots of water mass, therefore the third moment used for fitting will be much less. 3. Page 7, Line 3-14: Incomplete gamma distribution should be used here since only a limited range of particle size is used for fitting. I believe this is the reason why fitted Gamma DSDs are narrower (Page 8, Line 1-4) 4. Section 2.3. I have four major concerns for this method, and will elaborate them in next four points. As stated in Page 9, Line 9, this approach is suitable for the study of the same particle population, which is under Lagrangian framework. Therefore, aircraft dataset at different levels sampling different particle population cannot be used to track the change of cloud PSD gamma parameters, since they are not the same particle population. In addition, the PSDs at the same level are not the same and exhibit large variations. So, the best use of this technique will be for the parcel model if the authors can address the following three comments. 5. Even for the same PSD, there are large uncertainties as stated in Page 6, Line 27-Page 7, Line 1. McFarquhar et al. (2015) studied the uncertainties of counting statistics, and found that all the parameters within an ellipse in Gamma distribution parameter phase space are equally realizable. The displacement of gamma distribution parameters could be just random values in the ellipse unless the ellipse of equally realizable solutions are defined for each PSDs. 6. As for the "pseudo-forces", or microphysical processes which I prefer, this study decomposed it into two components: condensational growth and collision-coalescence growth. Due to the complex microphysical processes occurring in the clouds (as is discussed by the authors in Page 9, Line 21 – Page 10, Line 2), the evolutions of PSDs are very complex as some simulations using bin microphysics show. Simply relating a change of gamma distribution parameters to either condensational growth or collision-coalescence is not justified. Especially for any volume of air the aircraft sampled (or numerical models in Eulerian framework), the horizontal and vertical advection are very important. 7. The directions

and magnitudes of "condensational growth pseudo-force" and "collision-coalescence pseudo-force" are uncertain, which means that the influences of each individual microphysical processes on PSD evolutions are not studied clearly. The descriptions of "favors high value of mu while slightly increasing lambda" (Page 10, Line 4) and "lower values of lambda and mu, the former decreasing at a faster pace" (Page 10, Line 13-14) are not precise and not justified. The change of N_0 as described are wrong, since if condensational growth increase both $\mu$ and $\lambda$ while keeping the same total number concentration, N_0 should also increase. In addition, if collision coalescence lower both $\mu$ and $\lambda$, and total number concentration of course, then N_0 should be also decreasing. Besides, I would say that evaporation "pseudo-force" acts the opposite way as "condensational growth pseudo-force" instead of "collision-coalescence pseudo-force" in this study. Anyway, the directions of these "pseudo-forces" are totally unknown, and the change of gamma distribution parameters could be any microphysical processes since relating the change of gamma distribution parameters (or equivalently PSD moments or bulk properties) to any single microphysical process is impossible.

Minor comments: 1. Page 4, Line 24-25. This sentence needs to be elaborated. 2. Figure 1. Add flight height and temperatures for each flight. 3. Page 5, Line 23 – Page 6, Line 10. The three regions used in section 3.2 should be introduced here clearly. Furthermore, the cloud characteristic for coastal region and remote Amazon should be described here, similar to what has been written for the Arc of Deforestation. 4. Figure 5-8. The y and z axes ($\mu$ and $\lambda$) don't need to be taken logarithm for easy comparisons with previous studies. In addition, the projection of the 3D trajectories in N_0-$\mu$, N_0-$\lambda$, $\mu$-$\lambda$ planes will make readers to follow easier. 5. Figure 5-8. Add raw PSDs with different colors showing different time, so the change of PSDs is clear to the readers. As shown in many previous studies (e.g., Heymsfield et al. 2013), the gamma distribution parameters can compensate with each other, therefore, the different points in the gamma distribution parameter phase space could mean the same PSD. 6. Page 14, Line 23-27. Recommend removing these sentences. As stated in Major comment #7, the quantitative descriptions of these "pseudo-forces" are lacking.

Besides, the method may just work for Lagrangian framework. I cannot see how this could be used for bulk microphysical schemes. 7. Page 17, Line 10-21. According to Equation 9, this is similar to fix $\mu$ which is adopted in lots of numerical schemes. Actually, the small range of $\mu$ is due to its scale, and could mean large variations of PSDs. 8. Page 19, Line 11-12. The sentence that "The characteristics of the clouds warm layer. . .should have a determining role in the glaciation initiation". I would argue that the IN and the ice microphysics above are more important. The characteristic of IN between the remote Amazon and the Arc of Deforestation are not studied. The number concentration of ice particles above should also be analyzed, which may explain the differences in glaciation.

---

## Editor Comment (EC1) · A. Heymsfield (Editor) · 26 Sep 2017

My comments are attached

Please also note the supplement to this comment:
https://www.atmos-chem-phys-discuss.net/acp-2017-185/acp-2017-185-EC1-supplement.pdf

---

## Author Comment (AC5) · 3 Oct 2017

We would like to thank Dr. Andrew Heymsfield for the comments as Editor to this manuscript. Following are our responses, based on the supplement of your comment.

1) **(Comment)** Page 6, Line 7: The problem I see with the model is that no activation is allowed above cloud base. However, this has been observed in data I published for the NAMMA field program, in convective updrafts. Activation is through air introduced via entrainment aloft and also due to fallout of particles from parcels which therefore allows the supersaturation to build up.

1) **(Answer)** Yes, the model has several simplifications that affect the results. However, we believe that the lack of activation above cloud base is not of concern to this particular analysis. As we comment in Section 2.3 and in the supplement, we try to avoid new droplet formation because of its effects on the DSDs. Higher in the clouds, when the droplets are bigger, new droplet formation would produce bi-modal DSDs that would greatly affect the Gamma parameters. New droplet formation would result in wider Gamma DSDs that may not be representative of the situation. We believe the condensational growth with new droplet formation should be treated separately of the condensation exclusively on pre-existing droplets because they produce different patterns in the Gamma phase space. The same can be said about other processes such as entrainment (homogeneous or inhomogeneous), sedimentation, etc. We clarified in the fourth paragraph of Section 2.3 that the CCN activation takes place only at cloud base. Additionally, we removed the comment underlined in the following sentence of the supplement: "The vertical speed was fixed at 0.5 m s$^{-1}$ as we wanted to minimize the effect of new droplet formation in the DSD shape".

2) **(Comment)** Page 18, Line 2: I'm not terribly happy with this type of flight pattern-multiple convective clouds, some of which are different, are used in the analysis.

2) **(Answer)** We understand that the flight patterns may not seem ideal to the purposes of the type of analysis proposed here. However, the considerations used here bring us as close to the ideal scenario as possible within the limitations of aircraft campaigns.

3) **(Comment)** Page 20, Line 9: I would like to see how the addition of the CDP+CIP cloud drop spectra affect the gamma fit parameters. But, I totally agree that the sample volume for drizzle-size drops is very poor as is the sizing. This point should be made in the article.

3) **(Answer)** The addition of CIPgs would result in wider DSDs with bigger mean diameters, which affects the positioning in the Gamma phase space. In terms of the trajectories, it would obviously only affect the clouds that were able to produce rain – in our case mostly the cleaner clouds in RA1, RA2, and M1. Here we reproduce Figures 6 and 7 of the manuscript but only for the flights RA1 and AD2 and discriminating between "CDP-only" and "CDP+CIP" (using the underscore "prec" for the latter):

[Figure]

**Figure R1**: trajectories for flight RA1 for "CDP only" (RA1) and "CDP+CIP" (RA1$_{prec}$).

Note that for the flight AD2 (Figure R2), both trajectories are equal, given that there were basically no CIP-sized droplets. For flight RA1, the two trajectories start to deviate close to the 0 °C isotherm, and have big disparities in the mixed phase. That disparity is due to the formation of bigger hydrometeors that widen the DSDs and increases D$_{eff}$ (we did not discriminate between ice and liquid droplets in this figure). From Figure R1 we have an indication of the impacts of adding CIPgs to the DSDs, but we believe it is more valuable to focus exclusively on CDP.

[Figure]

**Figure R2**: trajectories for flight AD2 for "CDP only" (AD2) and "CDP+CIP" (AD2$_{prec}$).

We complemented the respective sentence in Section 2.2 (second paragraph) for clarity (new text is underlined): "The number of data with LWC$_{D>50}$ > 0.1 g m$^{-3}$ is only 12% of the number of DSDs with LWC$_{D<50}$ > 0.1 g m$^{-3}$, meaning that drizzle and precipitation are relatively infrequent in the dataset. This observation combined with the possibility of higher uncertainties (especially on the lower CIPgs bins) when combining two different instruments with distinct measurement principles further justify the focus exclusively on CDP". Please note that the percentage changed from 8% to 12%. I re-assessed the data and found that the 8% was wrong. That value was for a bigger dataset that included other flights, which did not make it to the final manuscript. By limiting for the flights mentioned in the paper, we obtained the new value of 12%. We apologize for the error and the text is now corrected.

4) **(Comment)** Page 20, Line 22: I wouldn't mind seeing a figure that compares the gamma fit parameters for the incomplete and complete gamma psd, at least for a few selected examples.

4) **(Answer)** There are studies in the literature that suggest that, even though DSD truncation have effects on the Gamma parameters, it may leave their inter-dependence relatively unchanged. For instance, Brandes et al. (2003) uses the μ-Λ relation found by Zhang et al. (2001) to constrain the Gamma DSD from dual-polarized radar retrievals:

$$\Lambda = 1.935 + 0.735\mu + 0.0365\mu^2$$

And they note that "Fitting the observations with a gamma or a truncated-gamma DSD has little effect on the µ-Λ relationship. Magnitudes for both parameters are proportionately smaller for a truncated DSD". Additionally, Ulbrich (1985) analyzed the impacts on rainfall integral parameters considering the DSD truncation. They found that the β exponent in relations between rainfall moments is relatively insensitive to truncation, in the form:

$$M_p = \alpha M_q^{\beta}$$

In other words, the moments may change but the relative relation between them remains the same. Given that the Gamma parameters are obtained from the moments in our manuscript, we expect a similar patter – Gamma parameters may change but the overall appearance of the trajectories remain the same. We added the following text to the paragraph right after Eq. (5):

"Previous studies comparing the complete and incomplete (or truncated) Gamma fits suggest that, while there are differences in the resulting parameters, the relation between them remains similar. The first indication of that comes from the study of Ulbrich (1985) that analyzed the relation between rainfall DSD moments in the empirical form $M_p = \alpha M_q^{\beta}$ where $p$ and $q$ are the two distinct moment orders and $\alpha$ and $\beta$ are fit parameters. The author notes that $\beta$ is relatively insensitive to DSD truncation, meaning that the relation between the moments remain similar while their overall values change. Brandes et al. (2003) also note that the µ-Λ relation introduced by Zhang et al. (2001) is relatively insensitive to DSD truncation. In the present study, the focus is more on the relation between the Gamma parameters rather than their values itself. For that reason, we favor the use of the complete Gamma".

Full references cited here:

Zhang, G., J. Vivekanandan, and E. Brandes, 2001: A method for estimating rain rate and drop size distribution from polarimetric radar measurements. *IEEE Trans. Geosci. Remote Sens.,* **39,** 830–841.

Zhang, G., Vivekanandan, J., Brandes, E. A., Meneghini, R., and Kozu, T.: The shape-slope relation in observed gamma raindrop size distributions: Statistical error or useful information?, Journal of Atmospheric and Oceanic Technology, 20, 1106–1119, doi:10.1175/1520-0426(2003)020<1106:TSRIOG>2.0.CO;2, 2003.

Ulbrich, C.W., 1985: The Effects of Drop Size Distribution Truncation on Rainfall Integral Parameters and Empirical Relations. *J. Climate Appl. Meteor.,* **24**, 580–590, https://doi.org/10.1175/1520-0450(1985)024<0580:TEODSD>2.0.CO;2.

5) **(Comment)** Page 21, Line 6: The Braga article is not yet accepted for publication and issues relevant to this reviewers' comment apply to the reviews of the Braga article as well.

5) **(Answer)** The paper is in fact published: https://www.atmos-chem-phys.net/17/7365/2017/. The confusion might come from the other Braga paper that is still under review: https://www.atmos-chem-phys-discuss.net/acp-2016-1155/.

6) **(Comment)** Page 22, Line 3: However, single turrets are not followed. Thus, only in a broad sense can the results be used.

6) **(Answer)** Yes.

7) **(Comment)** Page 25, Line 8: But multiple turrets are penetrated.

7) **(Answer)** Yes.

8) **(Comment)** Page 28, Line 19: As I mentioned earlier, there are sample volume and sizing issues for the CIP drizzle-size data.

8) **(Answer)** See our answer #3 here in this document.

---

## Author Response (AR1)

**Authors response to Anonymous Referee #1**

Major comments:

5  1. **(Comment)** This paper describes an attempt by the authors to use gamma function fitted using cloud macro- and microphysical observations to analyze microphysical growing path and particle size distribution evolution within deep convective clouds. The data used in this study are from ACRIDICON-CHUVA field campaign over Amazon, primarily six flights focusing on cloud microphysics measurements over regions with different aerosol background profiles. The findings of cloud properties under different environments appear to be very interesting. However, the method in using gamma function to interpret cloud microphysical growing path contains serious issues.

1. **(Answer)** We would like to thank Anonymous Referee #1 for the invaluable comments. Please find in this document the detailed answers to your concerns.

2. **(Comment)** The description of the closure of gamma function was firstly given by Eq. (2) - (4), where the closure variables were the zeroth, second, and third moment. The three undetermined parameters of gamma function would be defined by these moments. However, the actual closure variables, as described in Eq. (6) – (8) are liquid water content, cloud droplet number concentration, and effective droplet diameter. The former two were equivalent to the third and zeroth moment, respectively, while the effective droplet diameter was mostly equivalent to the first moment. To the least, these two descriptions are redundant. In fact, in many places of the paper including the Concluding remarks, the authors were still referred to the second moment. Indeed, the procedure of fitting gamma function with observations was never clearly described.

2. **(Answer)** The third paragraph and Equations (2) – (4) explain the methodology adopted to obtain the Gamma parameters. It is a relatively simple process and the method of moments has been extensively

studied in the literature. From the Gamma parameters obtained with this methodology, we are able to calculate any DSD property of interest - note that the Gamma parameters fully define the respective Gamma DSD. In this study, we were interested in analyzing (among other things) the parameters given in Equations (6) – (9). Those are not closure parameters to obtain the Gamma DSD, instead are properties

5   we obtain from them. Because we use moments of order 0, 2, and 3, $N_d$, LWC, and $D_{eff}$ obtained from the Gamma parameters will exactly match the observations. The relative dispersion, on the other hand, have dependency on the first moments and will be slightly different. However, as explained in the text after Equations (6) – (9), this parameters is still well represented by the Gamma fit used here.

10   3. **(Comment)** The most serious flaw of the proposed method exists in the procedure to interpret microphysics in the phase space of size distribution function. For a given air parcel, the ternary group of closure variables (mostly moments in different order) and undetermined parameters are bonded by mass conservation applied to the prognostic procedure of the former group, this defines the unique solution of both groups through the evolution of the air parcel, and they change accordingly due to the variations of the

15   closure variables induced by dynamical and microphysical processes. Note also that the closure variables must be conservative ones with well-defined sink and source besides advection and mixing terms. When fitting gamma function with multiple observations, however, one should realize that these observations are multiple snapshots likely represent different air mass origin either unmixed or mixed, therefore, they mostly reflect different ternary groups of the closure variables and hence their paths in the phase space

20   are irrelevant microphysically speaking unless a strong isentropic assumption (at least for any given horizontal plane) is made. This is why even in analyzing Eulerian modeling results, modelers usually derive microphysical and size distribution evolution within a parcel framework (can be conveniently derived from Eulerian grid parameters though), e.g., the "Twomey model". Only within such a framework does the analysis of size distribution evolution become meaningful.

3. **(Answer)** We understand your concern because we are not strictly using a Lagrangian measurement setup (which is not possible using airplanes). As there is no way to follow specific air parcels with aircraft measurements, we had to steer our analysis to get as close as possible to the cloud evolution. Firstly, the

flight patterns were specifically chosen as to measure growing convective clouds at their tops. Therefore, in each vertical step of the aircraft, we probed clouds that were "older" than the clouds probed on the last step (vertical steps were in ascending order). Additionally, the Gamma phase-space trajectories are only calculated for $w > 0$ m s$^{-1}$, making sure we only capture the growing part of the convective elements. The flight strategy is now clarified in Section 2.1, where we added the following sentence (second paragraph): "The latter step was deployed as follows. After the cloud base penetration, the aircraft performed several penetrations in vertical steps of several hundred meters. In each step, the aircraft penetrated the cloud tops available, thus avoiding precipitation from above. In this way, different clouds can be penetrated in the same altitude level and the vertical steps followed the growing cumuli field overall". This procedure was performed to allow interpreting the data (only w>0) as quasi-Lagrangian trajectories of the cloud parcels. We also performed simple calculations with the help of a Lagrangian parcel model. For details of the runs, please refer to our answers to Anonymous Referee #2 (item 2). Overall, the model calculations show that the condensation and collision-coalescence pseudo-forces act in similar directions to what we show in Figure 3. Therefore, we not only provide further justifications for Figure 3, but also show that the trajectories in Figures 5-8 can also be explained by the physical processes in the Lagrangian model. Of course, the Lagrangian model we chose is relatively simple given that it does not consider advection or turbulent mixing. On the other hand, it is capable of isolating the effects of condensation and collision-coalescence growth on the DSDs (bin microphysics) and, therefore, on the trajectories in the Gamma phase-space. The fact that we observed similarities between the Lagrangian trajectories and the observed ones is a strong argument in favor of our approach.

4. **(Comment)** By the way, many comments made by the authors are not accurate. For example, in the Abstract, the opening statement seems attempting to link our lack of understanding of the "tropical clouds" solely to the model representation issue of certain physical processes. The statement of "there is almost no study dedicated to understanding the phase space of this function…" is not accurate too. The properties of Gamma function along with many other probability distributions have been well studied and documented in statistics and applied mathematics literature. In the cloud physics and modeling field, the

evolution of conservative moments (mostly in the format of LWC, number concentration, and spectral disperse) have never been a rare topic in various mostly modeling studies.

The observations are invaluable for further our understanding of cloud physics and for evaluating models. Applying derivatives of these data, however, warrens special cautiousness.

4. **(Answer)** Yes, we agree that the phrasing regarding tropical clouds and their representation in models was not appropriate. The intended meaning is that, because of our lack of knowledge of tropical clouds, we still can't reproduce them adequately in models. We changed the first sentence in the Abstract to be: "The behavior of tropical clouds remains a major open scientific question, resulting in poor representation by models".

We also updated the Gamma function reference in the Abstract to: "However, even though the statistical characteristics of the Gamma parameters have been widely studied, there is almost no study dedicated to understanding the phase space of this function and the associated physics. This phase space can be defined by the three parameters that define the DSD intercept, shape, and curvature…".

**Authors response to Anonymous Referee #2**

Major comments:

1. **(Comment)** This manuscript introduces a technique for describing cloud processes using the drop size distribution gamma fit coefficients, and the trajectory of these coefficients in three-dimensional space. Comparisons within this phase space are made among clouds with different environmental conditions and linked to various cloud processes. While the manuscript is well written, I think some aspects of the paper need further work.

1. **(Answer)** We thank Anonymous Referee #2 for the invaluable comments. Please find in this document the detailed responses to your concerns.

2. (**Comment**) First, the physical insights that are provided are not closely linked to the coefficients them-
selves, and instead are reworked into pseudo-forces related to condensation and collision processes. How-
ever, the method used to decompose the trajectories into these pseudo forces is not clearly described, and
as a result I find it difficult to accept many of the explanations behind the patterns in the data.

2. (**Answer**) Indeed, the pseudo-forces presented are somewhat loosely defined, which is one of the main
reasons why we use "Illustration of microphysical processes…" in the title. The use of the Gamma phase
space as an entity is new to the microphysical studies and we do not aim to cover all its aspects in this
first introduction. We are already working in a new study focusing only on cloud modeling to extract the
10 pseudo-forces definition and to show how this approach can be useful for microphysical modeling. In this
study, our main interest is to show that we can study patterns in this space and that it can be useful to the
tools already implemented in models (or remote sensing applications) and to develop new ones.

That said, we followed your suggestion and dedicated efforts to better define and quantify the pseudo-
forces properties. We identified that the ideal tool to address this issue would be a relatively simple model
15 that solves the condensation and collision-coalescence growth using the bin approach instead of the bulk.
A model that fits those requirements is described in Feingold et al. (1999) – item "c" in section 3, where
we run only two parcels and not a bigger ensemble. This is a parcel model that treats the DSDs in 35
mass-doubling bins from 3.5 µm up to ~9 mm in diameter. The processes solved by the model are: 1)
CCN activation, considered to be composed of ammonium sulfate; 2) growth by condensation; 3) growth
20 by collision-coalescence and 4) effects of giant CCN on the DSD evolution (we turn this process off for
the purposes of this review). Other processes such as aqueous chemistry, complex aerosol composition,
trace gases and radiation (and the effects of those processes on the DSDs) are not treated. Additionally,
by being a parcel model, it does not consider turbulent mixing and sedimentation from above.

The characteristics of the model make it suitable to simulate the effects of condensation and collision-
25 coalescence growth in the DSDs, which we can use to show the related patterns in the Gamma phase-
space. We tried to produce results based on the conditions measured during flight AC09 (now RA1),
where we used the following parameters as input: 1) mean aerosol diameter $D_g = 1.55$ µm, with standard
deviation of 2.2 for the lognormal function of the aerosols; 2) pressure at cloud base of 890 hPa; and 3)

temperature at cloud base of 20.85 °C. The vertical speed was fixed at 0.5 m s$^{-1}$ as we wanted to minimize the effect of new droplet formation in the DSD shape. Under those conditions, we ran the model twice: one run with only condensational growth (CG run) and one with both condensation and collision-coalescence growth (C2G run). Both runs produced the exact same DSDs in the lower parts of the cloud where

5    the condensation dominates, but differed significantly when the collision-coalescence became active (around 1200 m, where cloud base is at 0 m). When the collision-coalescence process activates, $D_{eff} \approx 25$ µm and the condensational growth is much less effective. Therefore, it was possible to isolate both processes. Because there is no turbulent mixing or dilution with dry air, the droplet growth with altitude is much more pronounced in the model compared to our measurements during AC09. For this reason, we

10    do not limit the Gamma fit to $D < 50$ µm as in the paper. Otherwise, it would be difficult to capture the effects of the collision-coalescence process – droplets grow relatively quickly beyond the 50 µm mark. We fitted Gamma DSDs (using the same moments of order zero, two, and three as in the paper) to the model outputs every 20 seconds. Therefore, each point in the Gamma phase space represents the instantaneous DSD measured every 20 seconds. The results are shown in the following three figures.

15    Figure R1 shows the Gamma phase space for both runs, where "*" markers are related to CG run and squares to C2G. The arrows represent the displacement vector every 20 seconds, which is related to the respective pseudo-force (colors represent altitude above cloud base in m). Note that in the first 500 m the Gamma points are the same for both runs. This layer is defined by condensational growth alone and we observe a "zig-zag" pattern in the Gamma phase-space. When the trajectory is upwards in the "zig-zag",

20    they are similar to what we observed in the paper – that is, growing µ and Λ (and shrinking $N_0$) along with the condensational growth. On the other hand, the model results also show a downward (in the Gamma space) trend during condensational growth. We noted that when the trajectory is downwards, the Gamma fit does not represent the DSD width correctly. At those points, the fixed bins between 10 µm and 15 µm present fast-growing concentrations (when the droplets grow sufficiently to transition from

25    the lower bins) that disproportionately affects the Gamma DSD width. In the downward pattern, the Gamma DSD relative dispersion can be up to 150% higher than the binned DSD. When the process stabilizes, the trajectory returns to the upward trend and the Gamma and binned DSD widths get progressively closer (~20% to ~50% difference). Based in those results we can conclude that the condensational growth in the model produces trajectories in similar directions to what we observed in the paper.

[Figure]

**Figure R1**: Gamma phase-space for both CG and C2G runs. The "*" markers are relative to the CG run, while squares represent the C2G run. Arrows represent the displacement vector between each 20-s point, which is related to the respective pseudo-force. Colors represent altitude above cloud base in m.

10 Figure R2 shows the same points of Figure R1, but colored according to $D_{eff}$. Additionally, we show lines of constant $D_{eff}$ along a surface (not shown) of $N_d = 250$ cm$^{-3}$ similarly to Figure 10 in the paper. The lines start at 5 µm in the top and grow in 5 µm intervals up to 50 µm in the bottom line. When comparing the trajectories with the $D_{eff}$ lines, it is possible to see where the droplets are growing faster. For instance,

the condensational growth close to cloud base is very effective (because the droplets are smaller) and the trajectory tend to cross the $D_{eff}$ lines. However, when droplets reach $D_{eff} \approx 25$ µm, the trajectories get almost parallel to the lines, showing slower growth. On the other hand, the collisional growth accelerates with increasing $D_{eff}$. This is expected from theory, but it is interesting to quantify its effects on the spherical coordinates of the displacement vectors – Figure R3.

[Figure]

**Figure R2**: similar to Figure R1, but colored according to $D_{eff}$. The lines shown are lines of constant $D_{eff}$ along a surface of $N_d = 250$ cm$^{-3}$ as in Figure 10 in the paper, going from 5 µm (top line) to 50 µm (bottom line) – 5 µm intervals.

Figure R3 shows the spherical coordinates of the vectors in Figures R1 and R2. $\theta$ is the azimuth angle measured in the plane $\log(N_0)$ x $\log(\mu)$, being 0 at the $\log(N_0)$ axis and growing counter-clockwise. $\varphi$ is the elevation angle, measured from the plane $\log(N_0)$ x $\log(\mu)$ to the $\log(\Lambda)$ axis. The size of the vectors

is measured by r. In Figure R3 we excluded the points in the downward part of the "zig-zag" mentioned above. Non-filled circles in Figure R3 represent condensational growth alone, while filled markers represent collision-coalescence (colors are altitude above cloud base in m). It is possible to note that the elevation angle $\varphi$ is slightly positive for the condensational growth, decaying with $D_{eff}$. The average value of this angle is 0.26 °. It has small values mainly because of the bigger values of $\log(N_0)$ as compared to $\log(\Lambda)$. Nonetheless, the most important feature is its sign transition from condensational to collisional growth. On the latter, the angle seems to grow linearly with $D_{eff}$ (except for the last point) as the process intensifies – averaged value of -4.23 °. Overall, this angle is related to the DSD curvature trend – positive when the curvature is shrinking (condensational growth) and negative when the curvature is increasing (collisional growth).

The azimuth angle $\theta$ defines how $N_0$ and $\mu$ evolve along the trajectory. For the condensational growth, this angle averages 179.6 °, meaning growing $\log(\mu)$ and shrinking $\log(N_0)$. On the other hand, this angle averages -13.7 ° for collisional growth and results in the opposite trend for the parameters. Both observations are in line with what we observed in the paper – now there is at least some quantification of the angles. Note that the angles most likely have different values in our observations given the differences in the values of the Gamma parameters. However, their sign, and therefore the direction of the motion in the space, is the same between our model calculations and the observations shown in the paper. Finally, we can note that r tends to decrease as the condensation rates decay, but it does not increase as the collisional growth intensifies. However, the acceleration of the collisional growth is reflected in $\varphi$ and $\theta$ – both decrease, resulting in a trajectory that crosses the $D_{eff}$ lines in Figure R2.

Overall, the modeling results presented here clearly indicate that the patterns observed in the Gamma phase space in the paper are indeed related to the condensation and collision-coalescence processes. The relation between both processes and the evolution of the Gamma parameters are consistent between the Lagrangian simulation and the observations. The natural next step would be to calculate the speeds and accelerations (and therefore the actual pseudo-forces), but this will not be addressed in this introduction paper. The actual implementation of the concepts presented here would need further work that is beyond the scope of the present study. A study is underway using different parametrizations, aerosol properties and environmental properties.

We added three new paragraphs to Section 2.3 commenting on the Lagrangian model results and detailed it a little more in the supplement (with the figures/text shown here for the readers).

[Figure]

**Figure R3**: spherical coordinates of the displacement vectors shown in Figures R1 and R2. $\theta$ is the azimuth angle in the $\log(N_0) \times \log(\mu)$ plane, growing counter-clockwise (is 0 at the $\log(N_0)$ axis). $\varphi$ is the elevation angle from the $\log(N_0) \times \log(\mu)$ towards the $\log(\Lambda)$ axis and r is the size of the vectors. The colors represent altitude above cloud base in m.

3. **(Comment)** Secondly, gamma functions often provide good mathematical fits to drop size distributions, but attempting to understand cloud processes using the fit coefficients is fraught with difficulty, which I don't think is addressed sufficiently in this manuscript. Gamma function coefficients can vary

substantially depending on the fit method used, the size range over which the fit is made, and the suita-
bility of the underlying size distribution shape to be fit with a gamma. Many of these issues were ad-
dressed in the recent publication by McFarquhar et al. (JAS 2014). Using different fitting methods they
found that the N0 coefficient, for example, can vary by many orders of magnitude, even when the same
moments (1, 2, and 6) are used to make the fit. Using a different set of moments, like the 0th, 2nd, and
3rd used in this manuscript would likely result in even larger changes. Furthermore, the coefficients N0
and mu are inextricably linked, with N0 having the units of m^(-4-mu). So as mu changes, N0 will respond
mathematically, even though such a change may not represent a physical process.

3. **(Answer)** The Gamma function and its parameters are indeed complex to use in practical applications.
Additionally, the $N_0$, µ, and $\Lambda$ parameters can sometimes seem as abstract numbers that are mathemati-
cally loosely defined. In other words, those parameters can have extreme behaviors depending on the way
you choose to calculate them. However, their values and, perhaps most importantly, their interdependence
is singular in each methodology. For instance, we could have different values for the Gamma parameters
shown in the paper and the spherical coordinates shown in Figure R3 if we were to use, say, moments 3,
4, and 6 for the fit. If we were to compare between the two methodologies, it wouldn't be a fair comparison
because their internal functioning (i.e. their parameter space) is different. Fits that use higher-order mo-
ments have stronger weights for bigger droplets, affecting the parameters values and their phase space.
What we can do is to fix in a particular methodology and make the pattern analysis inside its particular
phase space. We specifically chose to use moments 0, 2, and 3 in order to obtain a parameter space that
is similar to what a bulk model should be able to reproduce. With regards to the moment method, we
believe this is the best approach given that it precisely reproduces at least 2 moments predicted by bulk
models (e.g. droplet number concentration and liquid water mixing ratio).

When the methodology is fixed, it doesn't really matter if, for instance, $N_0$ covers several orders of mag-
nitude. In our modeling calculations $N_0$ went from $\sim 10^{-150}$ to $\sim 10^1$, but all those values are inside the phase
space and can be expected when the DSDs fit certain criteria. We noted that $N_0$ reach such low values for
narrower DSDs, like the ones that appear after long periods of (exclusively) condensational growth.
Therefore, the theoretical phase space allows for such wide variability. The observations, on the other

hand, will of course cover a much more limited volume in the phase space. The idea is that both theoretical and observed phase spaces operate under the same underlying "laws" – at least considering only condensation and collision-coalescence growth, the model should be expanded to encompass other processes.

Regarding the linkage between $N_0$ and μ: as you correctly pointed out, those parameters are mathematically linked by definition. In fact, all three parameters are correlated in one way or another. When you go back to the equations used to obtain the parameters, this is very clear: first you obtain μ as a function of a dimensionless ratio between the moments, then you obtain Λ from μ, and finally $N_0$ from both of them. The relation between the Gamma parameters, modulated by the three moments, is a key aspect that generates the trajectories observed. If the parameters were completely independent, there wouldn't be trajectories in the phase space. There would probably be "clusters" of points for various types of DSDs. Our methodology aims to take advantage of this relationship in order to help on pattern recognition. It follows that the phase space is non-orthogonal, where it can "shrink" or "inflate" depending on the region of analysis. This is possibly one of the difficulties in applying this method to models, because the mathematic deductions are not straightforward. However, the ability to describe the microphysical evolution in this space opens new possibilities for DSD modeling, potentially improving subsequent calculations such as evaporation, sedimentation, etc.

I understand that the relationship between μ and $N_0$ is mathematical, but I would like to point out that it also makes physical sense. Take Equation 9 from the paper:

$$\varepsilon = \frac{\sigma}{D_g} = \frac{1}{\sqrt{\mu + 1}}$$

This equation states that the relative dispersion (or DSD width) can be calculated directly from μ. When μ increases, the DSD gets narrower. In that case, the left tail of the DSD gets closer to the maximum concentration diameter. Therefore, the intercept has to be lower and $N_0$ also shrinks. If you consider condensational growth, the situation is the same – see figures here and in the paper. Therefore, the linkage between $N_0$ and μ also have association to physical processes. When we look at the collision-coalescence growth, the opposite happens - μ decreases causing $N_0$ to increase.

4. **(Comment)** A more effective method may be to plot the moments themselves in 3D space rather than first fitting them to a gamma function. The moments are more easily linked to known microphysical processes, and if they are computed directly from the distributions do not suffer from the complications of poor fitting. The moments can always be computed from modeled DSDs as well, which would the avoid the further complications introduced when models use restricted gamma parameter spaces. At the very least, I think the authors should investigate the sensitivity of the observed phase-space patterns to different gamma fitting methods, and more clearly identify the source and interpretation of the F_cd and F_cl pseudo-forces.

4. **(Answer)** We believe that plotting the moments in a 3D space would be very similar to plotting the Gamma parameters. The moments are also not independent, thus resulting in a similar non-orthogonal space. However, the linkage between the moments-space and the underlying DSD is non-trivial. You would need to apply transformations to obtain the respective DSDs. Therefore, we believe the Gamma phase-space is more suitable in order to be portable for other applications. Also, as commented above, some calculations explicitly need the DSD parameters.

Regarding other methods to fit the DSDs, we tested a new fit based on moments of order 3, 4, and 6 (M346). Here we reproduce Figures R1-3 with this new approach (Figures R4-6). For this case, we had to limit the fittings to $D < 150\,\mu m$ because of the stronger weight to bigger droplets that caused negative values of $\mu$. Note that the patterns in the phase-space are very similar to the previous case (M023). Averaged values for $\theta$ and $\varphi$ are 179.5 ° and 0.35 ° for condensation and -19.4 ° and -4.0 ° for collision-coalescence, respectively.

[Figure]

**Figure R4**: same as Figure R1, but for M346.

[Figure]

**Figure R5**: same as Figure R2, but for M346.

[Figure]

**Figure R6**: same as Figure R3, but for M346.

5  Specific comments:

1. **(Comment)** Section 2.2: How were DSD shapes that are not well fit by a gamma function handled in the analysis (e.g. bimodal or skewed distributions)?

10  1. **(Answer)** No special routine was applied to bimodal or skewed distributions. The idea is to produce the phase-space of the observations as is (except for the filter to remove residual DSDs) within the limitations of the Gamma fit.

2. **(Comment)** Section 2.3: The introduction of the F_cd and F_cl pseudo-forces seem incomplete and leaves many unanswered question, such as: How were they determined, i.e. can they be presented mathematically? Do they completely describe the total force F? Are they orthogonal, if not, in which direction in the phase space does each force point?

2. **(Answer)** Please refer to our answer to your major comments and the new paragraphs in Section 2.3.

3. **(Comment)** Figures 3,5,6,7, and 8: It is difficult to determine where the lines and points are in 3-D space. A projection of the fit lines onto the X, Y, and Z planes would greatly help with the visualization.

3. **(Answer)** We left Figure 3 as is for simplicity, but added the requested projections to the other figures.

4. **(Comment)** Section 2.3: Given the sensitivity of N0 to the mu parameter, the speculations regarding N_0 would be much more convincing if N_d (or 0th moment) were used instead.

4. **(Answer)** Our affirmations about $N_0$ were not speculations, but based on our measurements – note that the trajectories shown in Figures 5-8 point out to the patterns commented in Section 2.3. The new model calculations also corroborate our affirmations.

5. **(Comment)** Section 3.2: The manuscript states that measurements were taken 'close to cloud top', but more information is needed about the placement of the measurements in the cloud. Was the aircraft making multiple passes to a fixed location, or attempting to intercept the same visual position in the cloud on mulitple passes? How long did the aircraft pattern take relative to the lifetime of the cloud, and at what point the life cycle of the cloud were the measurements taken?

5. **(Answer)** We added the following sentences to Section 2.1 to clarify the flight strategy: "The latter step was deployed as follows. After the cloud base penetration, the aircraft performed several penetrations

in vertical steps of several hundred meters. In each step, the aircraft penetrated the cloud tops available, thus avoiding precipitation from above. In this way, different clouds can be penetrated in the same altitude level, but the vertical steps followed the growing cumuli field overall".

6. **(Comment)** Section 3.3: How were the clean and polluted clouds determined? Were the flight patterns and environmental conditions for each of these clouds comparable?

6. **(Answer)** The clean and polluted clouds were determined based on the measurements shown in Table 1 – i.e. the aerosol concentrations. All flight patterns followed the steps we described in Section 2.1 and we also show them in the map in Figure 1. The environmental conditions are discussed in Figure 4 and Table 1.

**Authors response to Anonymous Referee #3**

Major comments:

**(Comment)** This paper uses fits of measured cloud droplet size distributions (DSDs) in gamma phase space to investigate warm-phase microphysical cloud properties and the role of "pseudo-forces" in affecting the evolution of the gamma parameters and the DSDs. Overall, I found the description of a unique set of data interesting and formative, and hence believe that the paper is worthy of publication. However, there are some issues that should be addressed in order to improve the presentation before the paper is published as discussed below.

**(Answer)** We would like to thank Anonymous Referee #3 for the invaluable comments. Below we address them individually as best we can.

1. **(Comment)** The authors segregate the flights that are flown into the different regions of the Amazon where they are flown. Although changes in surface conditions are no doubt important for affecting the

cloud properties, meteorological conditions can also have an important impact on cloud properties. Some comments about this should be added to the manuscript and some analysis of the meteorology on the different days should be added to see if such differences can also explain some of the variation in cloud properties. I think attributing much of the changes to aerosols is not fully justified until the meteorological context is further explored.

1. **(Answer)** The meteorological context is clearly determinant for cloud formation over the Amazon or over any other region. We address this issue in two steps in the paper. Firstly, we show satellite images in Figure 2, clearly indicating the predominance of cumuli fields for all flights chosen. Flight AC07 deviates a little from this pattern, where deeper convection was observed. However, the aircraft pilot usually avoided penetrating deeper convection for safety reasons (at least in the lower levels). Therefore, even if there was deeper convection in the region, the aircraft actually penetrated the growing convective elements around it. Additionally, we show temperature and humidity profiles in Figure 4. Those profiles show that, even though the cumuli fields are relatively similar, there are some thermodynamic differences. Clouds over the Southern Amazon were subject to drier and warmer air, justifying their higher cloud base altitude. We do not believe that those differences would significantly impact the characteristics of the warm-phase DSDs. They may impact the overall lifecycle of the clouds and precipitation, but the microphysics in the lower parts of the clouds likely depend on factors such as updraft speed and, most importantly, aerosol concentration and size distribution. The effects of thermodynamics and aerosol properties on cloud microphysics were studied in a precedent paper (same special issues - https://doi.org/10.5194/acp-17-10037-2017).

2. **(Comment)** I was a bit surprised on page 6 where the authors described that they were focusing on the CDP measurements where D < 50 micrometers. It would seem to me to be quite important to also examine the drizzle sized drops measured by the CIP, as when drizzle was present it would seem to be very important to account for that in the analysis. How are flights handled when there was some precipitation-sized drops? Were these larger drops incorporated into the analysis of simply ignored? Further, for Eq.

(2) to Eq. (4) should the incomplete gamma function rather than the gamma function be used to account for the fact that not the complete size range of particles were measured?

2. **(Answer)** In this study, we are really interested in analyzing cloud droplet physics rather than drizzle/precipitation physics. Those are rather different. For instance, the condensational pseudo force is mostly insignificant for precipitation-sized liquid droplets. We indeed removed the drizzle/precipitation droplets from CIP for the Gamma fit, similarly to what models do. Models separate cloud and precipitation DSDs, that can be combined if need be. Another reason is that droplets with D > 50 µm were relatively infrequent in our measurements. In the warm phase, T > 0 °C, the number of data with $LWC_{D>50}$ > 0.1 g m$^{-3}$ is only 8% of the cases where $LWC_{D<50}$ > 0.1 g m$^{-3}$ (i.e. only a small portion of the data contained significant amounts of drizzle/precipitation). We added the following sentences to the second paragraph in Section 2.2: "The intent is to focus on cloud droplet growth processes and bringing the analysis closer to modeling scenarios. Additionally, the percentage of data with significant liquid water content (LWC) for D > 50 µm is relatively small. The number of data with $LWC_{D>50}$ > 0.1 g m$^{-3}$ is only 8% of the number of DSDs with $LWC_{D<50}$ > 0.1 g m $^{-3}$".

Regarding the incomplete Gamma function, we reproduce our answer #4 for Anonymous Referee #4: "While we agree that the incomplete Gamma distribution would fit better the measurements, its use would result in other issues. As one of the main interests of the paper is to study the theoretical Gamma phase-space and its applicability to cloud modeling in the future, the use of the incomplete Gamma would not be ideal. In a modeling scenario, you don't have the observed DSD and therefore have no way of finding the truncation diameters. Additionally, the use of the incomplete Gamma distribution might add artificial patterns to the phase-space that are due to the truncation and not to physical processes".

3. **(Comment)** I think some more comments on the quality of the microphysical measurements are needed. How did the CAS and CDP probes compare? What are the estimated uncertainties in the size distributions? How did the LWC integrated from the CAS or CDP compare with bulk measurements from a hot-wire probe, which I am assuming were made. I am assuming that fits were only done to the liquid

distributions, or do you use all the distributions? This should be clearly stated when discussing the phase partitioning at the bottom of page 8.

3. **(Answer)** The instruments uncertainty, as well as their inter-comparison, is thoroughly analyzed in Braga et al. (2017) – same special issue (ACRIDICON-CHUVA). We updated the text with this reference (see second paragraph in Section 2.2). The fits were made to all measurements, irrespective of the NIXE-CAPS classification. However, this should not result in problems for our analysis given that the focus of the fits was primarily in the warm phase. Note that we do not draw any conclusions regarding the fits for regions above the transition between warm and mixed layers. We added the following sentence at the end on the NIXE-CAPS paragraph in page 8 to reflect this comment: "The NIXE-CAPS classification is a separate analysis and will not be considered as a filter to apply the Gamma fits to the CDP measurements. The CDP data fits are primarily focused on the warm phase and the transition to the mixed layer, where liquid droplets predominate".

4. **(Comment)** The implicit basis of the analysis presented in the Gamma phase space is that one is dealing with a Lagrangian case. But, inevitably, with any sort of microphysical measurements different samples of particle populations are being sampled. Further, there can be mixing and dynamic motions in clouds that would affect how the DSDs vary in the gamma phase space. Is there any way of representing these mixing processes on the diagram? I also think the action of the pseudo-forces and the impact of condensational growth and collision-coalescence could be better illustrated on the diagram. Can you show an example size distribution (it can be a theoretical rather than observed distribution if it is easier) and show how the size distribution would change due to simple model calculations of either condensational or collision-coalescence growth. Then, illustrate the location of all 3 DSDs (original, one undergoing condensational growth, and one undergoing collision-coalescence growth) on the gamma phase space and it will be easier for the reader to appreciate how these forces are represented on the diagram. Such simple theoretical/modeling calculations may also help you assess how the DSD characteristics are being affect by homogeneous/inhomogeneous mixing (discussion at top of page 12).

4. **(Answer)** As you correctly pointed out, it is impossible to produce Lagrangian trajectories based on aircraft microphysical measurements. Therefore, we have to make some assumptions to constrain our method. The flight patterns were specifically chosen in order to follow growing convective elements, where the aircraft penetrated the tops of the clouds. In this way, we both avoided precipitation from above

5 and also tightened the relationship between the altitude of the measurements and the lifecycle of the clouds. We do not presume to claim that this guarantees that our trajectories are Lagrangian. In fact, when it comes to the observations, we never mention it. We just use the altitude of the measurements as a proxy for cloud evolution, meaning that higher measurements present "older" droplets. We believe this is rather reasonable and is also common place in microphysical studies. The confusion might come from the way

10 we described the theory of the Gamma phase space in Section 2.3. In this idealized scenario, we can think of Lagrangian trajectories in order to facilitate the comprehension of the processes that affect DSD evolution. Now the link between the observations and the Lagrangian trajectories, for instance, should be addressed by other means such as modeling. As the title of the paper says, we illustrate the microphysical processes observed in the Gamma phase space rather than attempt to implement it in any actual modeling

15 tool. This is the natural next step, of course, which is already ongoing. We added the following sentence in the second paragraph of Section 2.1 to further detail the flight patterns and why they could be used as a proxy for cloud evolution: "The latter step was deployed as follows. After the cloud base penetration, the aircraft performed several penetrations in vertical steps of several hundred meters. In each step, the aircraft penetrated the cloud tops available, thus avoiding precipitation from above. In this way, different

20 clouds can be penetrated in the same altitude level, but the vertical steps followed the growing cumuli field overall".

In order to compare the results in the paper to a Lagrangian case, we ran a simple model. Please refer to the answer #2 to Anonymous Referee #2. From those calculations, we were able to conclude two things. Firstly, that the qualitative results from the model agree well with our observations. Therefore, even

25 though we could not produce Lagrangian observations, they agree with Lagrangian calculations. Secondly, we were able to test your suggestion regarding the actual calculations of the pseudo-forces (or at least displacements in the phase-space). We were able to confirm the overall directions of the pseudo-forces between the observations and the model, while also quantifying the displacements due to each

growth process. The details of the model run can be found in the mentioned answer #2 to Anonymous Referee #2. We also added three new paragraphs to Section 2.3 commenting on the Lagrangian results. Additionally, the answer to Anonymous Referee #2 were also compiled in a new supplement.

Regarding the effects of homogeneous or inhomogeneous mixing in the phase-space, we believe it is
5  beyond the scope of this work – which is focused primarily on condensation and collision-coalescence. However, it shouldn't be hard to analyze the effects on the phase-space. Note that there is significant literature regarding the effects of mixing on the DSDs. Therefore, one suggestion would be to apply the knowledge we have to a Gamma DSD and study the displacements in the phase-space.

10  5. **(Comment)** I'm wondering if some different terminology could be used to refer to the different flights. Although referring to flight numbers (e.g., AC19, AC18, AC12, etc.) might be very informative for people who were involved in the field program, I continually had to refer back to the table to remember the regions in which the flights were conducted to help interpret the data. Can you refer to them as maybe AC1 (AC19 for Atlantic coast 1), RA1 and RA2 (AC09 and AC18) for remote Amazon, and AD1, AD2
15  and AD3 (for AC07, AC12, and AC13) so that it is more easy to remember the flights going through the manuscript. Or, maybe some other terminology would also work.

5. **(Answer)** Indeed, this would greatly facilitate reading the paper. We kept the ACXX nomenclature in Table 1 for consistency with the other special issue papers and added the new definitions as: 1) M1 (as in
20  Maritime1 - AC1 might be confused with the ferry flight with the same nomenclature even though we don't mention it in our paper) for flight AC19; 2) RA1 and RA2 for AC09 and AC18 as you suggested; and 3) AD1, AD2, and AD3 also per your suggestion.

6. **(Comment)** With regards to the depiction of the DSDs in phase space, I would find it much easier is
25  some 2-d cross sections were presented in addition to the 3-d volumes (it was hard to follow some of the discussion on the contrasts between clean and polluted trajectories). It is very hard to visualize how the different parameters are changing on these 3-d plots, so some 2-d cross sections would also offer some

supplementary information. Further, what are the uncertainties or range of possible values in the gamma parameters.

6. **(Answer)** Thank you for this suggestion, we added the requested cross sections and it is much better now. With regards to the data spread around the trajectories, it can already be observed in Figures 5-7. Minor Comments:

1. **(Comment)** Page 2, Line 20: I was surprised to see that the undisturbed portions of the rainforest are said to have homogeneous surface properties: compared to oceanic surfaces surely the nature of the forest is somewhat inhomogeneous? On page 5 (lines 20-25), the authors talk about differences in surface and thermodynamic conditions on more of the disturbed areas of the Amazon, so I found that this comment was a bit misleading.

1. **(Answer)** Surface conditions over the forest is indeed less homogeneous than over the ocean. The intended meaning is to say that it is more homogeneous than urbanized regions. The sentenced was changed to reflect this: "…Given the relative homogeneity of the surface (as compared to urbanized regions) and the pristine air over undisturbed portions of the rainforest…".

2. **(Comment)** Page 3, Line 1: Typically the term ice nucleating particles (INPs) rather than ice nuclei (IN) now. See Vali (2015).

2. **(Answer)** Agree, thanks.

3. **(Comment)** Page 4, line 9: Unless specific numeric values are quoted, the parameters of the gamma function (or any parameter in general) do not have units associated with them. They could be given in any unit with an appropriate conversion being made. Recommend removing the units in parenthesis.

3. **(Answer)** Added a clarification explaining that the units given are the ones to be considered in this study.

4. **(Comment)** Page 5, line 23: if the convective clouds were growing, how could you ensure that the third stage was always flown through the growing tops? It would seem that different altitudes below cloud top might have been sampled for the different population of clouds.

4. **(Answer)** The aircraft performed several steps in altitude and in each level the pilot looked for cloud tops available for penetration. There may be some differences regarding the distance to the cloud top, but the overall intent was to minimize precipitation falling from above as much as possible.

5. **(Comment)** Page 7, Eq. (7). I don't think this equation is correct (the factor of 10^-9). Any equation must be unit-independent. Constants for conversions between specific unit sets hence don't belong in equations as those factors will automatically appear when converting between the different units of the variables.

5. **(Answer)** We are sticking to the units we actually used. We provide them right after the equations, therefore it is easier for the reader to understand directly what we did. Added the following sentence before the equations for clarification: "In the units considered here, the equations are given by:".

6. **(Comment)** Page 8, line 4: If the fact epsilon obtained by the gamma parameters does not match those from the DSDs suggest that the gamma distribution does not give a good fit to the DSD?

6. **(Answer)** We mention that the Gamma-epsilon and the Observed-epsilon are tightly linked by eps_gamma = 0.95*eps_obs ($R^2$ = 0.93). Therefore, it is safe to say that the observed epsilon is well represented by the Gamma fit. The Gamma DSD is only slightly narrower (angular coefficient of 0.95). Changed the sentence to: "The relative dispersion of the Gamma DSD may differ from the observations, given the differences between the parameterized and observed DSDs. However, our measurements show

that the Gamma and observed $\varepsilon$ are closely related by $\varepsilon_{Gamma} = 0.95\varepsilon_{Observed}$ ($R^2 = 0.93$), showing that the Gamma DSDs are slightly narrower on average" for clarification.

7. **(Comment)** Page 9, line 12: Can you use a different word rather than "phase transitions?" there is some confuse about whether you are talking about phase space or the phase (liquid, mixed or ice) of the cloud particles.

7. **(Answer)** Changed to phase-state transition.

8. **(Comment)** Page 11, line 16: "from drier air", can you list the humidities in Table 1?

8. **(Answer)** The humidity for the different regions is shown in Figure 4.

9. **(Comment)** Page 11, line 23: Do you mean average RH? Clouds do not form in an environment where the relative humidity at their location is between 60% and 90%. Can you also give some indication of the thickness of the different cloud layers?

9. **(Answer)** Added "surrounding environment" instead of only "environment" for clarification. This is the air around the clouds and not within them. It is hard to provide cloud thickness because they are growing as the airplane ascends. However, they can reach altitudes up to 15 km approximately.

10. **(Comment)** Page 12, lines 13-14: How do you know the observations were obtained close to cloud top? Unless you have remote sensing data or some ascents out of cloud, is it conceivable the particular cloud you were sampling extended to a higher height?

10. **(Answer)** As explained previously, we can be relatively sure that the aircraft penetrated the cloud top by the flight pattern planning. The pilot looked for cloud tops in each flight level.

11. **(Comment)** Page 13, line 1: What is classified as a significant difference? Was some sort of statistical test applied?

11. **(Answer)** Sentence changed to "At first glance, it is possible to see stronger differences between the trajectories in the different regions, while internal variations are much weaker".

12. **(Comment)** Page 14, line 12: What statistical test was applied to know that the res

12. **(Answer)** The text of this comment is cutout, so no changes were applied.

**Authors response to Anonymous Referee #4**

Major comments:

1. **(Comment)** This paper used in-situ data from six flights collected during the ACRIDICON-CHUVA field campaign to explore the linkage between gamma distribution parameter phase space and underlying microphysical processes. Three different environmental conditions, the Atlantic Coast, the remote Amazon, and the Arc of Deforestation were studied, and the differences in the underlying microphysical processes among these regions were compared. The paper fits into the scope of ACP and is generally well written, however, the approach used in this study has severe scientific flaws. Therefore, this paper needs to be revised considerably before it can be published in ACP.

1. **(Answer)** We would like to thank Anonymous Referee #4 for the invaluable comments. Please find in this document our detailed answers.

2. **(Comment)** Page 5, Line 15-18. Are there only six flights during the whole field campaign focusing on clouds? If not, why other flights are not used? Especially for Atlantic Coast, there is just one flight used.

2. **(Answer)** Yes, there were other flights that were partially dedicated to probe clouds. However, our specific focus on individual trajectories in this paper meant that we could exchange increased statistics for specialized analysis. The reasoning behind our flight selection was mostly due to the aerosol characteristics below clouds. All flights chosen presented relatively uniform aerosol number concentrations below clouds, therefore avoiding mixture of cleaner and more polluted clouds. On the other cloud profiling flights, we noted variations of the aerosol number concentrations, which would make the analysis much more difficult. By focusing on the selected flights, we were sure that each flight contained almost exclusively the same type of clouds in terms of aerosol conditions. The following sentence was added to the first paragraph in Section 2.1 for clarification: "There were other flights with cloud penetrations, but they are not considered in this study because of higher aerosol variability below clouds. The flights chosen for analysis presented relatively low aerosol variability, meaning that the clouds probed in the same flight were likely subject to similar aerosol conditions".

There was only one flight over the ocean, so we couldn't increase its statistics on the paper.

3. **(Comment)** Page 6, Line 22-24: Why PSDs from CIPgs is not used? Only using CDP to create PSDs with $D < 50$ um will miss out lots of water mass, therefore the third moment used for fitting will be much less.

3. **(Answer)** Our main interest in this study is on the Gamma trajectories of the cloud DSDs. We don't use CIPgs data because we consider them as drizzle/precipitation DSDs, therefore out of the scope of this study. Besides, there were relatively few DSDs for $D > 50$ µm. As we explained in our answer #2 to Anonymous Referee #3: "In the warm phase, $T > 0$ °C, the number of data with $LWC_{D>50} > 0.1$ g m$^{-3}$ is only 8% of the cases where $LWC_{D<50} > 0.1$ g m$^{-3}$ (i.e. only a small portion of the data contained significant amounts of drizzle/precipitation)".

4. **(Comment)** Page 7, Line 3-14: Incomplete gamma distribution should be used here since only a limited range of particle size is used for fitting. I believe this is the reason why fitted Gamma DSDs are narrower (Page 8, Line 1-4).

4. **(Answer)** While we agree that the incomplete Gamma distribution would fit better the measurements, its use would result in other issues. As one of the main interests of the paper is to study the theoretical Gamma phase-space and its applicability to models in the future, the use of the incomplete Gamma would not be ideal. In a modeling scenario, you don't have the observed DSD and therefore have no way of finding the truncation diameters. Additionally, the use of the incomplete Gamma distribution might add artificial patterns to the phase-space that are due to the truncation and not to physical processes. This point was clarified in the text right before Equation 2 with the following sentence: "The complete Gamma function is used to be consistent with modeling scenarios, where the Gamma parameters are calculated by:".

5. **(Comment)** Section 2.3. I have four major concerns for this method, and will elaborate them in next four points. As stated in Page 9, Line 9, this approach is suitable for the study of the same particle population, which is under Lagrangian framework. Therefore, aircraft dataset at different levels sampling different particle population cannot be used to track the change of cloud PSD gamma parameters, since they are not the same particle population. In addition, the PSDs at the same level are not the same and exhibit large variations. So, the best use of this technique will be for the parcel model if the authors can address the following three comments.

5. **(Answer)** Indeed, this is a valid concern that is also shared by the other reviewers. Here we reproduce our answer #4 to Anonymous Referee #3:

"As you correctly pointed out, it is impossible to produce Lagrangian trajectories based on aircraft microphysical measurements. Therefore, we have to make some assumptions to constrain our method. The flight patterns were specifically chosen in order to follow growing convective elements, where the aircraft penetrated the tops of the clouds. In this way, we both avoided precipitation from above and also tightened the relationship between the altitude of the measurements and the lifecycle of the clouds. We do not presume to claim that this guarantees that our trajectories are Lagrangian. In fact, when it comes to the observations, we never mention it. We just use the altitude of the measurements as a proxy for cloud

evolution, meaning that higher measurements present "older" droplets. We believe this is rather reasonable and is also common place in microphysical studies. The confusion might come from the way we described the theory of the Gamma phase space in Section 2.3. In this idealized scenario, we can think of Lagrangian trajectories in order to facilitate the comprehension of the processes that affect DSD evolution. Now the link between the observations and the Lagrangian trajectories, for instance, should be addressed by other means such as modeling. As the title of the paper says, we illustrate the microphysical processes observed in the Gamma phase space rather than attempt to implement it in any actual modeling tool. This is the natural next step, of course, which is already ongoing. We added the following sentence in the second paragraph of Section 2.1 to further detail the flight patterns and why they could be used as a proxy for cloud evolution: "The latter step was deployed as follows. After the cloud base penetration, the aircraft performed several penetrations in vertical steps of several hundred meters. In each step, the aircraft penetrated the cloud tops available, thus avoiding precipitation from above. In this way, different clouds can be penetrated in the same altitude level, but the vertical steps followed the growing cumuli field overall".

In order to compare the results in the paper to a Lagrangian case, we ran a simple model. Please refer to the answer #2 to Anonymous Referee #2. From those calculations, we were able to conclude two things. Firstly, that the qualitative results from the model agree well with our observations. Therefore, even though we could not produce Lagrangian observations, they agree with Lagrangian calculations. Secondly, we were able to test your suggestion regarding the actual calculations of the pseudo-forces (or at least displacements in the phase-space). We were able to confirm the overall directions of the pseudo-forces between the observations and the model, while also quantifying the displacements due to each growth process. The details of the model run can be found in the mentioned answer #2 to Anonymous Referee #2. We also added three new paragraphs to Section 2.3 commenting on the Lagrangian results. Additionally, the answer to Anonymous Referee #2 were also compiled in a new supplement".

6. **(Comment)** Even for the same PSD, there are large uncertainties as stated in Page 6, Line 27-Page 7, Line 1. McFarquhar et al. (2015) studied the uncertainties of counting statistics, and found that all the parameters within an ellipse in Gamma distribution parameter phase space are equally realizable. The

displacement of gamma distribution parameters could be just random values in the ellipse unless the ellipse of equally realizable solutions are defined for each PSDs.

6. (**Answer**) Yes, if we consider, for instance, the instrument uncertainty, we would end up with ellipsoids rather than points in the phase-space. However, given that the CDP uncertainty is about 10% (added this information to the second paragraph in Section 2.2), it is clear that the trajectories evolve beyond random movements in an ellipsoid. Let's consider, for the sake of this argument, that the instrument uncertainty is 10% for both the concentrations and the sizing of the droplets. In other words, let's consider that $N_d$, $D_{eff}$, and $D_g$ all have 10% uncertainty. Now how that would translate to the phase-space?

We visualize the situation in Figure 10 of the paper. Each uncertainty mentioned could be associated to one of the three axes of the ellipsoids, which are either tangent or normal to the $N_d$ surfaces in Figure 10. Consider that we are in the blue surface of Figure 10 ($N_d = 10$ cm$^{-3}$) and in the point where the blue trajectory crosses the upper black line where $D_{eff} = 23$ µm (they don't actually touch in the Figure 10, but let's consider they do). A 10% uncertainty in $N_d$ would mean a normal axis to the blue surface (both towards and away from the green surface). The size of this axis would be very small in the figure, being 1/100 of the distance between the blue and the green surfaces. The uncertainty in $D_{eff}$ would mean a tangent axis in the direction of the next black line (below), coincidentally of approximately the same size as the distance between the lines. If we consider that a 10% uncertainty in $D_g$ translates to 10% error in $\varepsilon$, then we have the last axis – also tangent to the blue surface, but in the same direction as the $D_{eff}$ line. The projection of the ellipsoid in the blue surface can be roughly represented by the purple curve in the figure below (note that the ellipsoid is very thin in the normal direction). Therefore, the trajectories cover wider regions than the ellipsoids dimensions and the trajectories approach is still valid. We added the following paragraph to the end of Section 2.3 to acknowledge the ellipsoid approach: "Another point to take into consideration are the ellipsoids discussed in McFarquhar et al. (2015). Basically, by considering the instrument and Gamma fitting uncertainties, it is possible to define volumes (with ellipsoid shapes) rather than individual points in the Gamma phase space. Inside each ellipsoid, all DSDs are equally realizable and therefore the movements within it have no particular physical meaning and are statistically the same. In this study, however, we estimate that the results evolve beyond individual ellipsoids and the patterns

are associated to physical processes. The results shown in the next sections will not consider the ellipsoid approach, but the points shown can be considered to be the central points of such volumes".

[Figure]

**Figure R1**: estimated size of the uncertainty ellipsoid in Figure 10 of the paper.

5   From this simple calculation, we conclude that the trajectories are likely to evolve beyond random movements in one ellipsoid. We won't carry this calculation over to the paper, however, because the trajectories can only be defined by points instead of volumes. In that sense, we can consider that the points in the trajectories are the central points of the ellipsoids. In the future, it would be interesting to study how the ellipsoids can be understood taking into account the underlying physics found in the phase-space.

7. **(Comment)** As for the "pseudo-forces", or microphysical processes which I prefer, this study decomposed it into two components: condensational growth and collision-coalescence growth. Due to the complex microphysical processes occurring in the clouds (as is discussed by the authors in Page 9, Line 21 – Page 10, Line 2), the evolutions of PSDs are very complex as some simulations using bin microphysics

15   show. Simply relating a change of gamma distribution parameters to either condensational growth or collision-coalescence is not justified. Especially for any volume of air the aircraft sampled (or numerical models in Eulerian framework), the horizontal and vertical advection are very important.

7. **(Answer)** Yes, there are several other processes that affect the DSDs. However, we would argue that condensation and collision-coalescence are definitely the most determinant. Any model, however simple, should be able to reproduce those mechanisms in order to explain precipitation formation even though the other processes are also important. These other forces are also being studied in the phase space using different microphysical parameterizations and we hope to have further results soon. For this first introduction of the phase-space, we chose to focus on condensation and collision-coalescence in order to define the overall characteristics of the space and how we can analyze cloud DSDs evolution in it. The other processes can and should be analyzed on top of that basis.

Let's take the mixing processes as an example. There are several studies analyzing the effects of homogeneous or inhomogeneous mixing on the characteristics of the DSDs. Therefore, in principle, we should be able to estimate the effects of mixing on Gamma DSDs as well, which can be reproduced in the phase-space. This could explain some characteristics of the trajectories we observed, but we believe this kind of analysis is beyond the scope of our paper. We are considering this kind of analysis as the necessary next steps, which should involve other tools such as models.

That said, we ran a simple Lagrangian model with bin microphysics in order to check the patterns associated to condensation and collision-coalescence and if they match our observations. If they qualitatively agree, it means that our observations are capturing those microphysical processes. Please refer to our answer #2 to Anonymous Referee #2 where we detail the model runs and its results. Overall, we were able to confirm that the condensation and collision-coalescence processes induce displacements in the same directions that we inferred from the observations. Obviously, the quantitative results are different given that the model is relatively simple and does not consider several processes that affect our observations. But we believe this is a good indication that the Gamma phase-space methodology is consistent and that we should dedicate efforts to progressively include the other processes as well. In that regard, it would be interesting to analyze the Gamma phase-space in more complex models such as the LES-type.

8. **(Comment)** The directions and magnitudes of "condensational growth pseudo-force" and "collision-coalescence pseudo-force" are uncertain, which means that the influences of each individual microphysical processes on PSD evolutions are not studied clearly.

8. **(Answer)** Yes, we didn't explicitly quantify the pseudo-forces in this first introduction of the methodology. The main intent of this paper is to introduce the Gamma phase-space as a physical entity and illustrate microphysical processes in it. We believe the quantifications should be the focus of future implementations of the phase-space and is beyond the scope of the present work. However, the Lagrangian model runs we mentioned in the previous item can be considered as the first step in that direction. Note that the direction of the displacements is similar between the model and our observations. Therefore, the displacements in the observed trajectories can be at least partially explained by the processes considered in the model.

9. **(Comment)** The descriptions of "favors high value of mu while slightly increasing lambda" (Page 10, Line 4) and "lower values of lambda and mu, the former decreasing at a faster pace" (Page 10, Line 13-14) are not precise and not justified. The change of N_0 as described are wrong, since if condensational growth increase both $\mu$ and $\lambda$ while keeping the same total number concentration, N_0 should also increase. In addition, if collision coalescence lower both $\mu$ and $\lambda$, and total number concentration of course, then N_0 should be also decreasing. Besides, I would say that evaporation "pseudo-force" acts the opposite way as "condensational growth pseudo-force" instead of "collision-coalescence pseudo-force" in this study. Anyway, the directions of these "pseudo-forces" are totally unknown, and the change of gamma distribution parameters could be any microphysical processes since relating the change of gamma distribution parameters (or equivalently PSD moments or bulk properties) to any single microphysical process is impossible.

9. **(Answer)** While our affirmations may not be precise, because it is pattern analysis instead of actual quantification, they are correct. Both our observations and the model calculations corroborate those affirmations. Please refer to our answer #2 to Anonymous Referee #2. In that document, we calculated the

elevation angle $\varphi$ to be positive for the condensational growth, meaning increasing $\Lambda$. On the other hand, we calculated an average $\theta$ of 179.6 °. Because this angle is calculated from the $\log(N_0)$ axis, this value means growing $\mu$ and shrinking $N_0$ – second quadrant. For collision-coalescence, $\varphi$ = -4.23 °, meaning decreasing $\Lambda$. The azimuth $\theta$ is -13.7 °, which is in the direction of growing $N_0$ and decreasing $\mu$. We

5  believe those quantifications are the first step at calculating the actual values of the pseudo-forces, but its implementation should be the focus of further studies in the future.

The "evaporation pseudo-force" is surely the opposite of the "condensational pseudo-force". But it is also true that the collision-coalescence pseudo-force acts in the opposite (overall) direction as the condensational pseudo-force.

Minor comments:

1. **(Comment)** Page 4, Line 24-25. This sentence needs to be elaborated.

15  1. **(Answer)** Changed to: "This process may produce artificial trajectories in the phase space by limiting the parameter variability".

2. **(Comment)** Figure 1. Add flight height and temperatures for each flight.

20  2. **(Answer)** We believe it would be hard to visualize the altitude and temperature in this figure. Instead, the requested properties can be seen in the other figures and tables in the paper.

3. **(Comment)** Page 5, Line 23 – Page 6, Line 10. The three regions used in section 3.2 should be introduced here clearly. Furthermore, the cloud characteristic for coastal region and remote Amazon should

25  be described here, similar to what has been written for the Arc of Deforestation.

3. **(Answer)** Added the following sentence to the end of the second paragraph in this section: "Contrasting with the Arc of Deforestation, the region named Remote Amazon in this study has much lower background aerosol concentrations, producing cleaner clouds. Clouds over the Atlantic Ocean developed under cleaner conditions as compared to the continental counterparts, and also had lower cloud bases (Table 1)".

4. **(Comment)** Figure 5-8. The y and z axes ($\mu$ and $\lambda$) don't need to be taken logarithm for easy comparisons with previous studies. In addition, the projection of the 3D trajectories in $N\_0$-$\mu$, $N\_0$-$\lambda$, $\mu$-$\lambda$ planes will make readers to follow easier.

4. **(Answer)** We analyzed the situation and decided that logarithmic axes were the easiest way to visualize the trajectories. The projections were added to the figures.

5. **(Comment)** Figure 5-8. Add raw PSDs with different colors showing different time, so the change of PSDs is clear to the readers. As shown in many previous studies (e.g., Heymsfield et al. 2013), the gamma distribution parameters can compensate with each other, therefore, the different points in the gamma distribution parameter phase space could mean the same PSD.

5. **(Answer)** We show observed and fitted DSDs in Figure 9. As far as the trajectories go, we did not observe any pair of similar DSDs with different Gamma parameters.

6. **(Comment)** Page 14, Line 23-27. Recommend removing these sentences. As stated in Major comment #7, the quantitative descriptions of these "pseudo-forces" are lacking. Besides, the method may just work for Lagrangian framework. I cannot see how this could be used for bulk microphysical schemes.

6. **(Answer)** We have shown that it is possible to produce quantifications of the processes using a simple Lagrangian model. Of course, the method should be refined to consider the many other aspects present in

clouds. We believe it is important to leave those affirmations in the paper, as they can be addressed by other researchers.

7. **(Comment)** Page 17, Line 10-21. According to Equation 9, this is similar to fix μ which is adopted in lots of numerical schemes. Actually, the small range of μ is due to its scale, and could mean large variations of PSDs.

7. **(Answer)** We left the second sentence in the last paragraph of Section 3.2 more open-ended: "If $\varepsilon$ can be constrained in the model, it should be possible to obtain the full Gamma DSD – which is the point in the intersection curve that presents the given $\varepsilon$".

8. **(Comment)** Page 19, Line 11-12. The sentence that "The characteristics of the clouds warm layer. . .should have a determining role in the glaciation initiation". I would argue that the IN and the ice microphysics above are more important. The characteristic of IN between the remote Amazon and the Arc of Deforestation are not studied. The number concentration of ice particles above should also be analyzed, which may explain the differences in glaciation.

8. **(Answer)** In this sentence we refer to the glaciation initiation – imagining a cloud that is growing past the 0 °C and does not have an ice phase yet. In this scenario, the characteristics of the droplets that cross the 0 °C isotherm are definitely important to trigger (or not) the glaciation process.
Yes, it is unsure how the IN population changes with pollution, but previous studies suggest that most of the IN over the Amazon come from natural sources – either from the forest or from long range transport (Saharan dust).

[revised manuscript text omitted]
 | $N_d$ | $LWC$ | $\varepsilon$ | $D_{eff}$ | $T$ | $UR$ | $w$ | *Adiabatic fraction* |
|---|---|---|---|---|---|---|---|---|---|

[revised manuscript text omitted]

---

## Author Response (AR2)

Authors response to Anonymous Referee #4

Anonymous Referee #4 comments:

1. **(Comment)** The authors added lots of new materials to address the comments made by four reviewers, and cleared some
of the concerns. The manuscript is improved significantly after revision. However, several major points raised in my previous
review are still not addressed well. In addition, the newly added Lagrangian model results are crucial to this research and it
would be nice to include part of them in the paper. Therefore, I recommend major revision.

1. **(Answer)** We thank Anonymous Referee #4 for the continued effort to improve this publication. Please find in this docu-
ment the description of how we addressed your concerns.

2. **(Comment - previous comment #4)** It would be really helpful to include a simple figure for comparison of fitting using full
and incomplete gamma distribution. The models are using integrated moments of gamma distribution doesn't guarantee
that full gamma distribution should be used for fitting of partially observed PSDs. I am not convinced that using incomplete
gamma distribution can always get similar values and relations of gamma distribution parameters. Just imagine fitting a PSD
without the mode diameter included as an extreme example…Adding a simple figure of comparisons will prove the validity
of the use in this paper if they do not make huge differences.

2. **(Answer)** We further analyzed the incomplete Gamma fit and its effects on the Gamma parameters. We can confirm that
the incomplete Gamma parameters have the same correlations among them by applying the method presented in Viveka-
nandan et al. (2004). The authors present a method to find the incomplete Gamma parameters by using an iterative process
for µ and Λ. They use moments of order 2, 4, and 6, but we adapt the methodology to match the moments used in the
manuscript (i.e. $0^{th}$, $2^{nd}$, and $3^{rd}$). The first step is to obtain the first guess using the complete Gamma – in that case, following

exactly the same methodology as in our manuscript. Secondly, the $G = \frac{M_2^3}{M_3^2 M_0}$ ratio is recalculated using the incomplete

Gamma function as in:

$$G_{inc} = \frac{[\gamma(\Lambda D_{max}, \mu+3) - \gamma(\Lambda D_{min}, \mu+3)]^3}{[\gamma(\Lambda D_{max}, \mu+1) - \gamma(\Lambda D_{min}, \mu+1)][\gamma(\Lambda D_{max}, \mu+4) - \gamma(\Lambda D_{min}, \mu+4)]^2} \tag{R1}$$

Where the "inc" subscript stands for "incomplete Gamma", $\gamma$ is the incomplete Gamma function, and $\Lambda D_{max}$ and $\Lambda D_{min}$ are
the intervals of integration. $D_{max}$ and $D_{min}$ were calculated from the measured DSDs as the largest and smallest bins with
droplet number concentrations higher than $10^{-6}$ cm$^{-3}$, respectively (including the CIPgs size range). When $G_{inc}$ is significantly
different than $G$ (we consider the significant threshold as 0.001 difference) it means the parameters should be adjusted in
order to produce the incomplete Gamma fit. In our dataset it was found that $G_{inc} \geq G$. When the difference $G_{inc} - G$ exceeds
the 0.001 threshold, we iterate µ and Λ until both ratios can be considered as equal. The iteration process is based on the
suggestions from Vivekanandan et al. (2004) and is as follows: for every step, µ is decreased by 0.01 and Λ is adjusted as:

$$\Lambda = \mu + 3\frac{M_2}{M_3} \qquad\qquad (R2)$$

Where $M_2$ and $M_3$ are the measured second and third moments, respectively. With the new $\mu$ and $\Lambda$ values, $G_{inc}$ can be recalculated by Eq. R1 and the process is repeated until $|G_{inc} - G| \leq 0.001$. After the final adjusted $\mu$ and $\Lambda$ values are found, the respective incomplete-Gamma-$N_0$ is calculated from $M_0$ as in Eq. (4) in the manuscript. For the dataset used in the manuscript, we found the incomplete and complete Gamma parameters differed in 64% of the cases. For those cases, $\mu$ and $\Lambda$ presented lower values while $N_0$ was higher. However, given that they were obtained by the same equations, the correlation between them remains the same. To better illustrate this point, we generated the two figures shown below. Firstly, we compare the range of the Gamma parameters that were adjusted or not by the method described above. The result is presented in Figure R1, where the title "Complete Gamma" refers to the complete Gamma fit as in the manuscript, while "Incomplete Gamma" refers to the cases where the iterative process was applied. We used only the 64% of the data where the complete and incomplete fits differed in both cases.

Figure R1 shows that the incomplete Gamma fit alters the DSD parameters only moderately. Shown in this figure are the 64% of the data with (right column) and without (left column) the incomplete Gamma adjustments. Regarding median values, $N_0$, $\mu$, and $\Lambda$ changed from $10^{-4.2}$ cm$^{-3}$μm$^{-1-\mu}$, 9.8, and 1.1 μm$^{-1}$ to $10^{-4.0}$ cm$^{-3}$μm$^{-1-\mu}$, 8.7, and 0.92 μm$^{-1}$, respectively. In terms of the overall histogram shape, we note relatively small variations in the modal values, where the differences are more pronounced in the tails of high $N_0$ and low $\mu$ and $\Lambda$.

The differences between the complete and incomplete Gamma fits can also be observed in the Gamma phase-space, as shown in Figure R2 (same 64% of the data as in Figure R1, limiting only to $D_{max} < 200$ μm). It is clear that the incomplete-Gamma-points are just shifted towards higher $N_0$ and lower $\mu$ and $\Lambda$ values, while their correlation remains intact because the same equations (apart from the iterative process) are used.

We believe that Figures R1-2 make it clear that the use of the incomplete Gamma may only change the parameters values and not their correlation. Therefore, the trajectories obtained later in the manuscript would be very similar with incomplete Gamma fits. We added significant explanations of this point after Eq. 5 in the manuscript. For simplicity, here we reproduce the text added:

"Previous studies comparing the complete and incomplete (or truncated) Gamma fits suggest that, while there are differences in the resulting parameters, the relation between them remains similar. The first indication of that comes from the study of Ulbrich (1985) that analyzed the relation between rainfall DSD moments in the empirical form $M_p = \alpha M_q^\beta$ where $p$ and $q$ are the two distinct moment orders and $\alpha$ and $\beta$ are fit parameters. The author notes that $\beta$ is relatively insensitive to DSD truncation, meaning that the relation between the moments remain similar while their overall values change. Brandes et al. (2003) also note that the $\mu$-$\Lambda$ relation introduced by Zhang et al. (2001) is relatively insensitive to DSD truncation.

In order to confirm that both the complete and incomplete Gamma fits result in similar correlations among the DSD parameters, a method similar to the one presented in Vivekanandan et al. (2004) was applied. This method aims at finding the incomplete Gamma parameters by using an iterative method to adjust $\mu$ and $\Lambda$. Here the moments of order zero, two, and three will be used instead of two, four, and six as in Vivekanandan et al. (2004). The first step is to calculate the ratio $G$ using the incomplete Gamma as:

$$G_{inc} = \frac{[\gamma(\Lambda D_{max},\mu+3)-\gamma(\Lambda D_{min},\mu+3)]^3}{[\gamma(\Lambda D_{max},\mu+1)-\gamma(\Lambda D_{min},\mu+1)][\gamma(\Lambda D_{max},\mu+4)-\gamma(\Lambda D_{min},\mu+4)]^2} \qquad (5b)$$

Where $\gamma$ is the incomplete Gamma function and $\Lambda D_{min}$ and $\Lambda D_{max}$ are its integration range. $D_{min}$ and $D_{max}$ were calculated from the measured DSDs as the lowest and highest diameter bins associated to drop concentrations higher than $10^{-6}$ cm$^{-3}$ (considering both CDP and CIPgs for testing purposes). This ratio is found to be greater or equal to its counterpart in Eq. 5. Therefore,

5   $\mu$ and $\Lambda$ should be lowered until $G_{inc}$ is sufficiently close to $G$ (the threshold of 0.001 is used here). For that purpose, $\mu$ was lowered in 0.01 steps where the respective $\Lambda$ is calculated by Eq. 3 until $G_{inc} - G \leq 0.001$. When this condition is met, $N_0$ is recalculated from Eq. 4.

For the incomplete Gamma fit, the adjustment described above was needed in 64% of the data used here. On the other cases, the ratios $G$ and $G_{inc}$ were similar and resulted in the same Gamma parameters. For the 64% of the dataset, median

10   relative differences in $log(N_0)$, $\mu$, and $\Lambda$ ranged from 10% to 20% towards higher $N_0$ and lower $\mu$ and $\Lambda$. However, regardless of their different values, the relation among the Gamma parameters remains unchanged in the incomplete fit because they are based on the same underlying equations. As will be shown later, the main interest of this study is in the relation among the Gamma parameters and not in their values itself. Therefore, we will focus on the complete Gamma noting that the results can be slightly shifted if truncation were to be considered".

[Figure]

**Figure R1**: histograms of the Gamma parameters adjusted (right column) or not (left column) by the iterative processes involved in the incomplete Gamma fit.

[Figure]

**Figure R2**: effects of the incomplete Gamma fit on the Gamma parameters in the phase-space.

3. **(Comment - previous comment #6)** The equally-realizable volume of gamma distribution parameters may be added in Fig.
10 for a single example point to support the conclusion that the trajectories cover a much wider region. The volume may
5   also partly quantify the uncertainty of the "pseudo forces" the authors want to illustrate.

3. **(Answer)** We added the suggested ellipsoid to Figure 10. The figure now is the same as the one we added to question #6
of your previous review. We added the following new text to the third paragraph in Section 3.3 to explain the new figure:

"The purple ellipse represents an estimative of the size of the ellipsoids introduced by McFarquhar et al. (2015). For the
calculation of the ellipse, we considered that the 10% error in the CDP measurements translates to 10% error in $N_d$, $D_{eff}$, and
10   $\varepsilon$. Under those conditions, the size of the ellipsoids can be represented by the purple ellipse in Figure 10. Note that its di-
mension in the normal direction from the $N_d$ surfaces is very small (1/100 of the distance between the surfaces shown), that
is why we only show an ellipse rather than an ellipsoid. This simple estimate is meant to find the order of magnitude of the
ellipsoids of McFarquhar et al. (2015) and they show that the trajectories are not just random displacement inside one ellip-
soid (the trajectories evolve beyond them)".

4. **(Comment - previous comments #7, #8 and the newly added materials in response to reviewer #2)** The "pseudo-forces",
or microphysical processes as I prefer, definitely change the PSDs and their moments (e.g., number concentration, mass
content, etc.). How exactly they change the PSD gamma distribution parameters, which are quantitive descriptions of PSDs,
are the novel part of this paper. Therefore, I recommend compiling the Lagrangian model results into Fig. 3 (or later) to
5  present the effects of "pseudo-force". The field campaign datasets may not be ideal to examine the linkage between gamma-
distribution phase space and "pseudo-forces" as several reviewers worried, but the Lagrangian modeling greatly helps for
the illustration purpose. By the way, the bin-scheme simulated PSDs may not be shaped like gamma distribution. The authors
are welcome to present several simulated PSDs and fitted PSDs to disapprove me.

4. **(Answer)** We really don't want to focus on modeling in this publication. The main reason is that the team is working
10 specifically on that issue right now. Two manuscripts are being prepared using column model. One studies the effects of
different parameterizations and aerosol characteristics (mean volume diameter, concentration and hygroscopicity) on the
trajectories, comparing with the observed trajectory presented in this study. It is interesting to see how well the bin micro-
physics is coherent to the observed data and how different is the two-moment parameterization. The second manuscript is
devoted to studying the pseudo-forces and quantitatively discussing them throughout the cloud life cycle. Therefore, we
15 believe it is better to have a separate and more thought-out publication that can cover more aspects of the Gamma phase-
space in modeling scenarios than to keep adding addendums to the current publication. We understand that the goal of this
study is to introduce the concept of trajectories in the Gamma space, which is the reason why we use the wording "Illustra-
tion of…" in the title. The follow-up studies are also the reason why we decided to submit the Lagrangian results as supple-
ment material. Hopefully the supplement material will be enough to bridge the gap between the publications in the near
20 future. We intent to submit these two new studies in the same special issue.

Regarding your question about the shape of the DSDs predicted by the Lagrangian model, we generated Figure R3. This figure
shows two DSDs and their Gamma counterparts. The first DSD ($DSD_1$ and the Gamma counterpart $\Gamma DSD_1$) is located at 100
m above cloud base and is mainly associated to condensational growth. $DSD_2$ (and $\Gamma DSD_2$) is higher in the cloud – 700 m
above cloud base - where collision-coalescence is already acting. Note that, although the Gamma fit is not perfect, the mod-
25 eled and fit DSDs do present similar shapes.

[Figure]

**Figure R3**: example comparison between DSDs generated by the Lagrangian model (continuous lines) and their Gamma counterparts (dashed lines).

[revised manuscript text omitted]